# Mechanisms of northern North Atlantic biomass variability

Galen A. McKinley[1,2], Alexis L. Ritzer[1], and Nicole S. Lovenduski[3]

[1]Department of Atmospheric and Oceanic Sciences, University of Wisconsin—Madison, Wisconsin, USA
[2]now at Columbia University and Lamont Doherty Earth Observatory, New York, USA
[3]Department of Atmospheric and Oceanic Sciences and Institute of Arctic and Alpine Research, University of Colorado Boulder, Colorado, USA
*Correspondence to*: Galen A. McKinley (mckinley@ldeo.columbia.edu)

**Abstract.** In the North Atlantic Ocean north of 40 $^{o}$N, intense biological productivity occurs to form the base
of a highly productive marine food web. SeaWiFS satellite observations indicate trends of biomass in this region over 1998-2007. Significant biomass increases occur in the northwest subpolar gyre and there are simultaneous significant declines to the east of 30-35 $^{o}$W. These short-term changes, attributable to internal variability, offer an opportunity to explore the mechanisms of the coupled physical-biogeochemical system. We use a regional biogeochemical model that captures the observed changes for this exploration. Biomass
increases in the northwest are due to a weakening of the subpolar gyre and associated shoaling of mixed layers that relieves light limitation. Biomass declines to the east of 30-35 $^{o}$W are due to reduced horizontal convergence of phosphate. This reduced convergence is attributable to declines in vertical phosphate supply in the regions of deepest winter mixing that lie to the west of 30-35 $^{o}$W. Over the full timeframe of the model experiment, 1949-2009, variability of both horizontal and vertical phosphate supply drive variability in
biomass on the northeastern flank of the subtropical gyre. In the northeast subpolar gyre horizontal fluxes drive biomass variability for both timeframes. Though physically-driven changes in nutrient supply or light availability are the ultimate drivers of biomass changes, clear mechanistic links between biomass and standard physical variables or climate indices remain largely elusive.

## 1 Introduction

Surface ocean phytoplankton contribute 50% of global net primary productivity [Field et al. 1998], form the base of the oceanic food web, and contribute to ocean sequestration of carbon dioxide [Sarmiento and Gruber, 2006]. The North Atlantic north of 40 °N experiences a strong annual cycle of productivity that is controlled by the interplay of physical and biogeochemical processes.

In general terms, marine phytoplankton growth is limited by nutrients in the subtropics and by light at subpolar latitudes [Fay and McKinley, 2017]. In the subtropics, an enhanced bloom occurs with relief of nutrient stress when vertical mixing is enhanced. In contrast, subpolar regions should have a reduced bloom with enhanced mixing because mixing enhances light limitation. Sverdrup [1953] used observations from a weather ship in the Norwegian Sea to propose the notion of a "critical depth" for subpolar regions. When the mixed layer reaches below the critical depth, physical mixing cycles phytoplankton through dark regions at depth which increases light limitation and decreases production. Dutkiewicz et al. [2001] and Follows and Dutkiewicz [2002] directly characterize productivity drivers with the ratio of the spring critical depth to the winter mixed layer depth in a theoretical model and compare to observations. Their relationships most accurately represent satellite and in situ observations in the North Atlantic subtropics and are less predictive in the subpolar gyre. Also identified is an intergyre region where observed relationships do not fit this conceptual model, presumably because both nutrient and light limitation are of first-order importance.

In recent decades, ocean color satellites have allowed for synoptic assessments of surface ocean productivity and its variability [Yoder and Kennelly, 2003; McClain et al., 2004; Siegel et al., 2005]. The first few years of data from the satellite Sea-viewing Wide Field-of-view Sensor (SeaWiFS) indicated that the seasonal cycle of productivity is largely consistent with the "critical depth" hypothesis [Siegel et al. 2002]. More recently, there has been an active debate about whether ecological processes may be more important to the subpolar spring bloom than the relief of light limitation due to mixed layer shoaling, as proposed by Sverdrup. Behrenfeld [2010, 2014] use satellite records to argue that phytoplankton accumulation is most significant in winter due to mixing that dilutes their interaction with grazers and other drivers of loss, and further, that the spring bloom does not represent a significant change in biomass accumulation rates. These findings are supported by analysis of an ocean model [Behrenfeld et al. 2013]. In situ observations using autonomous platforms, however, continue to support the conclusion that the springtime shoaling of mixed layers that relieves light limitation is coincident with a substantial increase in the rate phytoplankton biomass accumulation [Mahadevan et al. 2012, Mignot et al. 2018]. The physical mechanisms most important for springtime shoaling remain in discussion [Taylor and Ferrari, 2011, Mahadevan et al. 2012].

Longer records of ocean color reveal large-scale interannual changes in ocean productivity. Explanations for multi-year changes, and by extension expected future trends with climate warming [Bopp et al., 2013], tend to be based on the conceptual model of vertical processes controlling either nutrient or light limitation. Multiple analyses suggest that increased large-scale middle and low latitude stratification due to ocean warming limits the vertical supply of nutrients to the surface ocean and thus causes reductions in productivity

[Behrenfeld et al. 2006; Polovina et al. 2008; Martinez et al. 2009]. However, in the North Atlantic and the North Pacific subtropics, it has been found that local interannual variability in stratification is uncorrelated with local productivity, findings that do not support a one-dimensional mixing-productivity framework [Lozier et al. 2011, Dave and Lozier 2010, 2013]. Instead, it has been shown that large-scale correlations between chlorophyll and sea surface temperature (SST, a proxy for stratification) at low and mid latitudes can be explained by advective processes in the equatorial Pacific [Dave and Lozier, 2015]. At subpolar and polar latitudes, Behrenfeld et al. [2016] find that satellite-based estimates of imbalances between phytoplankton growth and loss can drive biomass interannual variability. Yet the fundamental importance of Behrenfeld's proposed ecological mechanism remains in debate both for seasonal and interannual timescales [Hunter-Cevera et al. 2016, Mignot et al. 2018].

In sum, there is growing evidence that the modification of light and nutrient limitation by vertical processes is alone insufficient to explain observed variability in surface ocean productivity. At the same time, there is growing evidence that horizontal physical processes could play a role, particularly in the northern subtropical gyre or "intergyre" region of the North Atlantic [Williams and Follows, 1998; Dutkiewicz et al. 2001; Follows and Dutkiewicz 2002; Oschlies 2002; McGillicuddy et al. 2003; Dave et al. 2015].

Williams and Follows [1998] illustrate that on the mean, horizontal Ekman fluxes are critical to surface nutrient supply in the North Atlantic from 40-60 $^{o}$N. However, Williams et al. [2000] find variability of horizontal fluxes to be an order of magnitude smaller than convective flux variability, and thus conclude that vertical processes dominate anomalies. Considering deeper processes, Williams et al. [2006] compare the magnitude of Ekman upwelling to the three-dimensional movement of volume or nutrients from the permanent thermocline to the full mixed layer, or "induction". Climatologically, nutrient supply to the subpolar gyre by induction is many times larger than the supply by Ekman upwelling. Induction is how the "nutrient stream" [Pelegrí et al., 1996; Palter et al., 2005; Williams et al., 2006] is accessed to allow for large-scale supply of nutrients from outside to inside the subpolar gyre. To our knowledge, interannual variability in induction has not been discussed in the literature. Further consideration of both horizontal and vertical processes is warranted with respect to understanding of temporal variability in surface ocean productivity in the North Atlantic.

Changing ocean circulation should influence horizontal and vertical transports of nutrients in the northern North Atlantic. A slowdown of the gyre should relax isopycnal slopes and decrease geostrophic advection along isopycnals. The North Atlantic subpolar gyre has exhibited substantial change since the 1950s when regular observations began to be available [Lozier et al. 2008]. There is evidence these changes occur in response to changing buoyancy forcing and wind stress, in turn associated with modes of climate variability, specifically the North Atlantic Oscillation and East Atlantic Pattern [Häkkinen and Rhines, 2004; Hátún et al., 2005; Lozier et al., 2008; Foukal and Lozier, 2017]. Via Ekman processes, reduction in wind stress should directly reduce upwelling in the subpolar gyre and also the horizontal transport of nutrients [Williams et al. 2000; Dave et al. 2015]. Buoyancy and turbulent fluxes also impact mixed layer depths and influence bloom

timing and strength [Bennington et al. 2009]. Consistent with this expectation, links between physical changes in the subpolar gyre and in situ observed changes in nutrients and ecosystems at several subpolar timeseries sites have been suggested [Johnson et al. 2013, Hátún et al. 2016, 2017].

In this study, we use a regional model to illustrate how changing light limitation and changing vertical and horizontal nutrient supply led to the significant changes in surface ocean biomass that were observed by SeaWiFS over 1998-2007 in the North Atlantic north of $40^o$ N (Figure 1c). This is a mechanistic analysis of the drivers of SeaWiFS-observed changes in biomass that are best quantified as linear trends given the 10 year prime observational period. The degree to which these drivers are responsible for internal variability

across the full model experiment (1948-2009) is also explored. Our approach can be contrasted to other possible approaches such as the use of empirical orthogonal function (EOFs) to consider dominant modes of variability [Ullman et al. 2009; Breeden and McKinley 2016]. The negative of EOF-type analysis is that it tends to explain at most 30% of the large-scale variance, and thus does not fully explain observations. This paper is a case study in which we aim to explain the drivers of the observed changes as fully as possible using

a model that represents well the observed changes.

## 2 Methods

### 2.1 Satellite data

Our analysis focuses on the period 1998-2007. Monthly SeaWiFS data becomes inconsistent beginning in

2008. For study of interannual trends, avoiding the need to fill gaps in the record is desirable. For additional comparison and extension of the record, biomass estimated from MODIS for 2003-2015 is also presented, again selecting years for which all months are available. For both SeaWiFS and MODIS, biomass is estimated using the updated CbPM algorithm [Westberry et al. 2008]. Additionally, we compare trends of modeled net primary productivity (NPP) to NPP from SeaWiFS estimated with both CbPM and the VGPM

algorithms (Behrenfeld and Falkowski, 1997). All data were provided by the Ocean Productivity Group at Oregon State University (http://www.science.oregonstate.edu/ocean.productivity/index.php, SeaWiFS biomass downloaded 11.28.16; MODIS biomass downloaded 01.24.18; NPP downloaded 05.29.18).

### 2.2 Regional hindcast model

The Massachusetts Institute of Technology General Circulation Model configured for the North Atlantic

(MITgcm.NA) [Marshall et al., 1997a; Marshall et al., 1997b], is used. The model domain extends from 20 °S to 81.5 °N, with a horizontal resolution of 0.5° x 0.5° and a vertical resolution of 23 levels that have a thickness of 10 m at the surface and gradually become coarser to 500 m thickness intervals for depth levels deeper than 2200 m. NCEP/NCAR Reanalysis I daily wind, heat, freshwater, and radiation fields from 1948-2009 force the model [Kalnay et al., 1996]. To correct for uncertainties in air-sea fluxes, SST and SSS (sea

surface salinity) are relaxed to monthly historical SST [Had1SSTv1.0, Rayner et al., 2003] and climatological

SSS [Antonov et al. 2006] observations, on the timescale of 2 and 4 weeks, respectively [Ullman et al. 2009].

To characterize sub-grid-scale processes, the Gent-McWilliams [Gent and McWilliams, 1990] eddy

parameterization, the KPP boundary layer mixing schemes [Large et al., 1994], and Fox-Kemper et al. [2008]

submesoscale physical parameterization are used. The phosphorus-based ecosystem is parameterized

following Dutkiewicz et al. [2005], and with modest revisions by Bennington et al. [2009]. This ecosystem

has one zooplankton class and two phytoplankton classes ("large" diatoms and "small" phytoplankton). The

biogeochemical model explicitly cycles phosphorus, silica and iron, and complete carbon chemistry is also

included. This model is identical to the one presented in Breeden and McKinley [2016], and uses the same

biogeochemical code as Bennington et al. [2009], Ullman et al. [2009] and Koch et al. [2009].

The coupled model has previously been shown to capture the timing and magnitude of the subpolar spring

bloom chlorophyll and its variability as observed by SeaWiFS [Bennington et al. 2009]. Mixed layer depths,

carbon system variables and nutrients are well simulated at Bermuda and in the northwest subpolar gyre

[Ullman et al. 2009; Koch et al. 2009]. As is common to this type of moderate-resolution model, productivity

in the subtropics is too low [Bennington et al. 2009]. Physical variability since 1948 is consistent with

observations [Breeden and McKinley, 2016].

As in Breeden and McKinley [2016], the physical model was spun up for a 100 year period with 1948-1987

repeated twice and then followed again by 1948-1967, for a total physical spin up of 120 years. The

biogeochemical model was then initialized using World Ocean Atlas phosphate concentrations and spun up

for 10 years using 1948-1957 daily forcing. To avoid initialization shock, the model was then forced for 5

years with repeating 1948 fields before the 1948-2009 experiment started. Due to Had1SSTv1.0 fields only

being available through 2009, this model integration ends in 2009. Future studies using Had1SSTv1.1, which

extends beyond 2009, will require re-initialization and new spin up integrations.

**2.3 Phosphate diagnostics**

To assess the processes modifying phosphate concentration, we employ phosphate diagnostics that quantify

flux convergences (in mmol m$^{-3}$ yr$^{-1}$) for net biological processes, vertical advection and diffusion, and

horizontal advection and diffusion. These terms describe the tendency of each process at every time step

during the model simulation, averaged to monthly for output [Ullman et al., 2009; Breeden and McKinley,

2016]. For conciseness, the biological uptake term presented here is the sum of separate diagnostic terms for

phosphate utilization by primary producers and remineralization that returns phosphate to the water column.

For analysis of mean and linear trends for 1998-2007, we use biological, vertical and horizontal diagnostic

terms. Unfortunately, the biological diagnostics prior to 1998 were lost after simulations were completed.

Thus, for correlations for 1949-2009, we use biomass in place of the biological diagnostics. This choice is

supported by strong correlations (R=-0.87 to -0.98) between biomass and the biological diagnostics in our

three focus regions (defined below) for 1998-2007. Biomass and biological diagnostics have an opposite sign because phosphate is removed as biomass accumulates.

## 2.4 Light and nutrient limitation

As detailed in Dutkiewicz et al. [2005], model phytoplankton growth is limited by light and the most limiting nutrient. Limiting nutrients are phosphate ($PO_4$) and iron (Fe) for small phytoplankton and $PO_4$, Fe and silicate ($SiOH_4$) for large phytoplankton. There is no nitrogen cycle in the model, consistent with other ecological models of comparable complexity (Galbraith et al. 2010). The parameterization uses Michaelis-Menton ratios that tend to 0 as the resource becomes severely limiting to growth, and approach 1 when replete. A lower value indicates a greater stress, and thus the phytoplankton group with the larger half saturation constant will be more limited for the same ambient nutrient or light concentration.

Specifically, maximum growth rates ($\mu_{max,small}$ = 1/1.3 $d^{-1}$, $\mu_{max,large}$ = 1/1.1 $d^{-1}$) are reduced through multiplication by limitation terms. $T_{func}$ modifies maximum growth based on temperature following Eppley (1972).

$$\mu = \mu_{max} \cdot T_{func} \cdot \gamma_{light} \cdot \min(\gamma_{PO4}, \gamma_{Fe}, \gamma_{siOH4(large\ only)}) \tag{1}$$

With half saturation constants $I_{o,small}$ = 15 $Wm^{-2}$, $I_{o,large}$ = 12 $Wm^{-2}$, light limitation is:

$$\gamma_{light} = \frac{I}{I+I_o} \tag{2}$$

And for nutrients

$$\gamma_X = \frac{X}{X+K_{o,X}} \tag{3}$$

For phosphate, X = $PO_4$ and $K_{o,PO4,small}$ = 0.05 mmol $m^{-3}$ and $K_{o,PO4,large}$ = 0.1 mmol $m^{-3}$. For iron, X = Fe and $K_{o,Fe,small}$ = 0.01 μmol $m^{-3}$, $K_{o,Fe,large}$ = 0.05 μmol $m^{-3}$. For large phytoplankton only, silicate limitation also applies, with $K_{o,SiOH4,large}$ = 2 mmol $m^{-3}$. Because of their higher half saturation constant for phosphate, modeled large phytoplankton are more phosphate stressed than small phytoplankton. In contrast, the higher light half saturation makes small phytoplankton experience greater light stress. Due to high levels of aeolian dust deposition in the North Atlantic, parameterized here with the imposition of climatological fields from Mahowald et al. [2003], iron is never limiting in our study area and is not further discussed.

For this analysis, monthly mean light and nutrient fields are used to calculate limitation terms for light and nutrient for each phytoplankton type.

## 2.5 Analysis

Throughout the study, annual averages over the top 100 m are used. This depth is selected because it is a reasonable approximation for both the euphotic zone and the Ekman layer, and is a computationally efficient choice consistent with previous work [Long et al. 2013; Williams et al. 2000, 2014]. For analysis of light limitation, however, it is important to consider that deep mixing will move mixed layer phytoplankton to substantially below 100 m [Sverdrup, 1953]. This effect would be poorly captured if light limitation terms were averaged only over the surface 100 m. The more appropriate choice, used here, is to use either the depth of the monthly mixed layer or 100 m, whichever is deeper. Light limitation is calculated monthly in this way and then annually averaged. For consistency, we apply the same averaging approach for nutrient limitation. However, since nutrients are homogenized by deep mixing, results for nutrient limitation are not substantially different from this calculation using of a strict 100m average. Limitation terms are not biomass-weighted.

For physical comparisons, mixed layer depth (MLD) is calculated using monthly density fields and a criteria of 0.03 kg m$^{-3}$ increase above the surface density. The barotropic streamfunction is calculated using a north to south integration of the full depth zonal velocity fields [Breeden and McKinley, 2016]. To find the minimum barotropic streamfunction of the subpolar gyre, the minimum within a region 60-30 $^{\circ}$W, 50-65 $^{\circ}$N is used. A preliminary comparison of nutrient flux variability to climate indices uses the winter (DJFM) East Atlantic Pattern (http://www.cpc.ncep.noaa.gov/data/teledoc/ea.shtml, downloaded 12.15.2017) and the winter North Atlantic Oscillation (Hurrell and NCAR, 2017).

This analysis is based on annual mean fields for both the observations and the model. A 3-month lag of the biology diagnostics and biomass fields after physical diagnostics and other physical fields is employed to account for the maximum physical forcing occurring in the winter prior to the spring bloom. Thus, annual mean physical fields are averaged from October of the prior year to September of the year in question. The use of 0, 1, 2 or 4 month lags leads to lower correlations, but does not substantially modify results. Biological fields are January to December averages.

To compare directly to the 10-year period of prime SeaWiFS observations, our primary focus is on linear trends over 1998-2007, with significance bounds set at $p < 0.05$ (95%). To complement this analysis with a consideration of interannual variability across the full model experiment (1948-2009), we also consider correlations of physical and biogeochemical timeseries calculated as area-weighted averages over three selected regions (defined below), and then linearly detrended prior to correlation analysis. Because of the aforementioned biological lag, the timeframe for correlations becomes 1949-2009.

**3 Results**

**3.1 Model comparison to observations**

The simulation captures the magnitude of mean 1998-2007 subpolar biomass reasonably in comparison to the satellite-based observations (Fig. 1a, b). The detailed spatial pattern of biomass is impacted by the North Atlantic Current extension being too diffuse and too directly east-west (i.e. not turning to the northeast as it

should at about 25 °W), as is common in models of this resolution [Williams et al. 2014]. The maximum of biomass is displaced to the east. Also, subtropical biomass is too high in the Gulf Stream extension, but otherwise too low in the remainder of the basin and, thus the gradient from south to north in the model from 35-50 °N is too sharp [Bennington et al. 2009].

Despite these imperfections, the model captures well the pattern and magnitude of statistically significant biomass trends north of 40 °N over 1998-2007 (Fig. 1c, d). In both observations and the model, biomass declines to the east of 30 °W from 40-50 °N and 35 °W from 50-60 °N, while it increases to the west. For simplicity, we refer to this boundary as 30-35 °W in our discussion.  Model trends are slightly weaker than the observed trends, but the coherent regions of statistically significant change are of similar size. Declines

to the east occur in two regions in both model and observations, one in the northeast and one in the southeast. Consistent with the mean biomass structure, simulated biomass trends are not in exactly the same locations as observed, but are displaced about 5° to the south in the southeast and northwest, and 5° to the south and 5° west for the northeast region. Comparison to net primary productivity (NPP) from SeaWiFS estimated with both the CbPM algorithm and older VGPM algorithm indicate comparable changes as in biomass, though

trends in the northeast are not significant for NPP (Fig. S1).

In both observations and models, the magnitudes of these changes are large in comparison to the mean. In the declining regions where mean biomass is 15-25 mgC m$^{-3}$ (Fig. 1, 2, S2), trends of -0.5 to -1.5 mgC m$^{-3}$ yr$^{-1}$ over 10 years lead to changes of 30-50%. In the increasing region to the west, changes are of similar magnitude. To focus our analysis, we select three regions in the model that capture these significant biomass

changes (Fig. 1d). We will use these regions for discussion and for averaging of biogeochemical and physical terms. In the northeastern subtropical gyre, or "intergyre" [Follows and Dutkiewicz, 2002], lies our southeast (SE) region, just south of the physical separation between the subpolar and subtropical gyre based on the barotropic streamfunction (section 3.4). The SE region is bounded between 30-15 °W and 40-50 °N. The northeast (NE) region lies in the eastern subpolar gyre to the southeast of Iceland, 35-20 °W and 55-60 °N.

The northwest (NW) region is south of Greenland at 35-20 °W and 50-60 °N. Regional mean changes in biomass from SeaWiFS (in the model) are -19% (-17%) in the SE region and -15% (-10%) in the NE.  To the west of 30-35 °W in the NW region, regional mean changes are +6%  (+9%).

In these three regions, annual anomalies of simulated biomass are compared to estimates from the SeaWiFS (1998-2007) and MODIS (2003-2015) satellites (Fig. 2). Monthly biomass timeseries are presented in Fig.

S1. In all regions, simulated biomass anomalies are quantitatively different from the observations to a similar degree that the observations differ from each other. In the SE region, the shift from positive biomass anomalies before 2004 to negative anomalies after 2004 is found in SeaWiFS and the model, and MODIS indicates a return to positive anomalies after 2010 (Fig. 2a). Of the three regions, this is the one where annual changes in the model are significantly correlated to those is SeaWiFS (R = 0.79, p < 0.05), despite the small

sample size (n = 10).  Linear trends for 1998-2017 in SeaWiFS and the model are the same, -0.41 mgC m$^{-3}$ yr$^{-1}$; however, the observations are better explained by this trend (R$^2$ = 0.76 for SeaWiFS, R$^2$ = 0.35 for

model). In the NE region, higher frequency variability is suggested, with mostly positive anomalies over 1998-2003 and negative anomalies from 2005-2009 (Fig. 2b). In this region, the linear trend for 1998-2007 in SeaWiFS is -0.26 mgC m$^{-3}$ yr$^{-1}$ (R$^2$ = 0.15) and -0.34 mgC m$^{-3}$ yr$^{-1}$ (R$^2$ = 0.50) in the model. The spatial displacement between the modeled and observed anomalies (Fig. 1c, d) is not accounted for with the regions used for Fig. 2b, but these comparisons do not substantially change if the averaging region for the observations in the NE is shifted 5 degrees north and 5 degrees east (not shown). In the NW region, positive anomalies of comparable magnitude dominate 2003-2008, the timeframe over which the three records coincide (Fig. 2c). Negative anomalies are largely found both before and after. In the last 3 MODIS years, positive anomalies return to the NW region. For 1998-2007, the linear trend in SeaWiFS is +0.22 mgC m$^{-3}$ yr$^{-1}$ (R$^2$ = 0.22) and for the model +0.23 mgC m$^{-3}$ yr$^{-1}$ (R$^2$ = 0.55). Having demonstrated that this model reasonably captures the patterns and magnitudes of biomass change, we now use the model to explain the mechanistic drivers in all three regions over the SeaWiFS period, 1998-2007.

### 3.2 Nutrient changes

Modeled anomalies are not due to zooplankton top-down pressure on biomass, as evidenced by zooplankton trends that are positively correlated with biomass trends (Fig. S3). Thus nutrient and light, the bottom-up drivers in this model that change in a manner that drives biomass changes consistent with observations (Fig. 1), are the focus of this analysis. The model captures the large-scale pattern of the phosphate field well, but mean values are 10-20% too low across most of the subpolar gyre (Fig 3a, b). Changes in the nutrient field could drive these observed and modeled changes, and as temporally resolved large-scale nutrient datasets are not available, the model alone allows us to evaluate nutrient trends (Fig 3c). Modeled nutrient concentrations decline significantly over 1998-2007 across most of the region north of 50 $^{o}$N. The pattern suggests these changes are important to the declines of biomass in the SE and NE regions. However, there is no increase of phosphate in the NW region where biomass was observed to increase.

### 3.3 Trends of light and nutrient limitation

To better understand drivers of simulated biomass trends, a next step is to decompose the biomass trends into those occurring in the small and the large phytoplankton (Fig. 4). On the mean, in the open waters of the North Atlantic, simulated large phytoplankton have a greater contribution to the total biomass in the north and west (Fig 4a), while small phytoplankton are dominant to biomass throughout the basin and particularly in the south and east (Fig 4b). Trends in simulated small phytoplankton contribute most to total biomass change (Fig 1d) in the SE and NW regions, while large phytoplankton trends are more important in the NE region (Fig. 4c, d).

Phosphate limitation for large phytoplankton has a strong gradient of more limiting in the south and east to less limiting in the northwest (Fig. 5a), while light limitation for small phytoplankton has largely a south to north gradient from less to more limiting (Fig. 5b). Trends over 1998-2007 in the model limitation terms illustrate that the SE and NE declines of simulated biomass are spatially coherent with enhanced phosphate

limitation (Fig. 5c), while the NW increase in biomass is spatially coherent with regions experiencing relief of light limitation (Fig. 5d). As shown in Fig. S2, mean and trends for light limitation for large phytoplankton and phosphate limitation for small phytoplankton have nearly identical patterns.

This distinction between the dominant limitations driving simulated biomass change in the east and west is borne out with detrended interannual correlations over the full model period, 1949-2009 (Table 1). In the SE region, phosphate limitation is strongly correlated with both small ($R_{SE\ (small,\ PO4)}$ = 0.68) and large phytoplankton ($R_{SE\ (large,\ PO4)}$ = 0.74), while light limitation is anti-correlated with biomass, i.e. less biomass occurs with more light, clearly illustrating that light is not the driving limitation. With respect to limitation

terms in the NE region, the only significant correlation for large phytoplankton is to nutrient limitation ($R_{NE\ (large,\ PO4)}$ = 0.31). Thus, the large phytoplankton that quantitatively dominate the 1998-2007 biomass decline (Fig. 4d) due to nutrient limitation (Fig. 5c) also vary by a similar mechanism over the full model experiment.

In the NW region, the 1998-2007 biomass trend is dominated by small phytoplankton (Fig. 4d) via light limitation (Fig. 5d), and these relationships also hold for the full model experiment. Small phytoplankton

light limitation is positively correlated with small phytoplankton biomass ($R_{NW(small,light)}$ = 0.61), while small phytoplankton biomass is reduced when more phosphate is available ($R_{NW(small,PO4)}$ = -0.50, Table 1). Though large phytoplankton have the opposite sensitivities ($R_{NW(large,light)}$ = -0.63, $R_{NW(large,PO4)}$ = 0.83), they are a smaller portion (40%) of the total biomass (Fig. 4a,b) and have a lesser role in total biomass changes (Fig. 4c).

In the model, silicate is also limiting to large phytoplankton and its limitation also becomes more intense over 1998-2007 to the north of 50 $^o$N (Fig. S4f). Yet, variability in silicate limitation is highly correlated to variability of phosphate limitation in the NW and NE areas for 1949-2009 ($R_{NE(PO4,SiOH4)}$ = 0.91, $R_{NW(PO4,SiOH4)}$ = 0.83, Table S1). Due to these high correlations and the fact that large phytoplankton are only dominant to biomass trends in the NE region (Fig. 4c), the remaining analysis addresses only phosphate fluxes. For

completeness, 1998-2007 light and silicate limitation trends for large phytoplankton and phosphate limitation for small phytoplankton are shown in Fig. S2, and 1949-2009 correlations in the three regions are given Table S1.

**3.4 Physical changes and their impacts on light and nutrient limitation**

Significant physical changes in the subpolar gyre influence the simulated nutrient and light fields. The model

barotropic streamfunction experiences a positive change from a minimum value of -41 Sv for 1998-2000 to -28 Sv for 2005-2007 (Fig. 6, Fig. S5). With this anomaly, the zero line of the streamfunction shifts several degrees north at 45-40 $^o$W and more modestly to the north in the east (bold black contours in Fig. 6a and Fig. 6b). The North Atlantic Current (NAC) flows along this contour, indicating a northward shift of the NAC.

Consistent with the weakening of the subpolar gyre, mixed layers shoal substantially, particularly to the west

of 30-35 $^o$W (Fig. 7a, b). A dramatic shoaling of maximum mixed layers is found in the NW region, going

from almost 1200 m to less than 400 m (Fig. 7e, Våge et al. 2008). This shoaling explains the strong decline in light limitation in the NW region. There is modest shoaling of mixed layers in the NE region (Fig. 7d) and there is no significant trend in the SE region (Fig. 7c). Shoaling in the NE could contribute to the reduction in phosphate availability and reduced biomass. However, the lack of mixed layer depth change in the SE
suggests that less vertical mixing is not the dominant driver of reduced biomass here.

**3.5 Phosphate diagnostic analysis**

To fully assess the three-dimensional physical drivers of phosphate supply to the NE and SE regions, we employ the phosphate diagnostics that quantify flux convergences. On the mean for 1998-2007 across the northern North Atlantic, vertical advection and diffusion supply phosphate to the euphotic zone (Fig. 8a),
with the supply being much stronger in the region of deepest mixed layers (Fig. 7). Horizontal advection and diffusion strongly diverges the converging vertical flux (Fig. 8b), leading to strong negative fluxes (divergence) coincident with strongly positive vertical fluxes. The horizontal flux divergence centered at about 30-35 $^{o}$W leads to positive horizontal fluxes (convergence) to the east and west. As the sum of the vertical and horizontal components is net positive (Fig. 8c), the mean three-dimensional advection and
diffusion net supplies phosphate to the subpolar gyre. The pattern of this supply is strongly influenced by both vertical and horizontal processes. Biological processes remove the physically-supplied phosphate from the surface ocean (Fig. 8d).

To east of 30-35 $^{o}$W, horizontal and vertical phosphate flux convergences are comparable in magnitude and both supply nutrients to the surface (Fig. 8, 9). In the SE region, mean 1998-2007 vertical advection and
diffusion supplies 0.13 mmol m$^{-3}$ yr$^{-1}$ while horizontal supplies 0.07 mmol m$^{-3}$ yr$^{-1}$, together supporting biological utilization of -0.20 mmol m$^{-3}$ yr$^{-1}$ (Fig. 9a). In the NE region, the mean vertical supply is 0.24 mmol m$^{-3}$ yr$^{-1}$ while horizontal supplies 0.08 mmol m$^{-3}$ yr$^{-1}$, and thus a mean biological utilization of -0.32 mmol m$^{-3}$ yr$^{-1}$is supported (Fig. 9b). The net convergence of vertical and horizontal fluxes in these regions can be contrasted to the NW region where the mean vertical flux converges 0.33 mmol m$^{-3}$ yr$^{-1}$, and from this
the horizontal flux diverges about 25% (-0.08 mmol m$^{-3}$ yr$^{-1}$) and biology diverges the remainder (-0.24 mmol m$^{-3}$ yr$^{-1}$, Fig. 9c). In all three regions, note that the variability of both horizontal and vertical flux convergences are of magnitudes comparable to the biological flux variability (Fig. 9).

For 1998-2007, simulated trends in the supply and removal of phosphate indicate a large decrease in supply via vertical fluxes to the west of 30-35 °W (Fig. 10a) and a corresponding strong reduction in the horizontal
divergence of phosphate, a positive anomaly (Fig 10b, 9c). To the west of 30-35 $^{o}$W, these opposing trends of the vertical and horizontal flux convergence largely negate each other. However, to the east of 30-35 $^{o}$W there are weak and mostly negative tendencies in the vertical and significant negative trends in the horizontal terms. Thus, the net physical phosphate supply in the eastern subpolar gyre has an overall negative trend, albeit only large enough to be formally statistically significant in parts of the SE region (Fig. 10c). The pattern
of reduced phosphate supply is consistent with the pattern of significant reduction in biomass (Fig. 1d) and

significant positive tendencies in the biological diagnostic indicating reduced phosphate utilization by phytoplankton (Fig. 10d). In summary, the model indicates that from 1997 to 2008, reduced vertical convergence of nutrients to the west of 30-35 $^{o}$W led to less horizontal convergence of nutrients to the east of 30-35 $^{o}$W, and thus less phosphate available for biological production. In this model, this mechanism is

sufficient to explain biomass changes consistent with SeaWiFS-observed declines in biomass in the eastern subpolar gyre (NE region) and northeastern subtropical gyre (SE region).

Long-term (1949-2009) correlations between physical diagnostic terms and biomass support the conclusion that variability in horizontal flux convergence is important to modeled biomass interannual variability to the east of 30-35 $^{o}$W (Table 2).  In both the SE and NE regions, horizontal flux convergence is significantly

correlated to biomass ($R_{SE(Biomass,Horiz)}$ = 0.44, $R_{NE(Biomass,Horiz)}$ = 0.48), suggesting that the 1998-2007 relationships are indicative of interannual behavior over the long-term, wherein reduced horizontal nutrient convergence leads to reduced biomass. For the SE region on the long term, vertical fluxes have also a significant correlation ($R_{SE(Biomass,Vert)}$ = 0.63), indicating that longer term interannual change in biomass in this region is determined by variability in both horizontal and vertical flux convergences. In the NW region,

biomass and horizontal flux convergence is also positively correlated ($R_{NW(Biomass,Horiz)}$ = 0.69), but this appears to be an indirect relationship. As light limitation is relieved, biomass increases, and at the same time vertical convergence of phosphate is reduced (Fig 10a) and there is a positive anomaly in the horizontal divergence (Fig. 10b). Consistent with this interpretation, vertical and horizontal convergence are strongly anti-correlated in the NW region ($R_{NW(Horiz,Vert)}$ = -0.76).

The impact of the physical drivers discussed earlier in this section on 1949-2009 biomass variability varies by study region. For the SE region, where biomass is positively driven by both horizontal and vertical nutrient flux convergence, biomass declines with positive anomalies of the minimum barotropic streamfunction of the subpolar gyre ($R_{SE(Biomass,PsiMin)}$ = -0.37, Table 2), consistent with 1997-2008 relationships. However, the minimum barotropic streamfunction is not itself correlated to either horizontal or vertical flux convergence.

Vertical supply is not a significant driver for 1997 to 2008 changes, but for the longer term, vertical nutrient fluxes decrease with shallower mixed layers (a negative anomaly) and warmer temperatures ($R_{SE(MLD,Vert)}$ = 0.66, $R_{SE(SST,Vert)}$ = -0.64).

For both the NE and NW regions, correlations between changing minimum barotropic streamfunction (Fig. 6), shoaling mixed layers (Fig. 7) and horizontal and vertical nutrient convergence (Fig. 10) for 1949-2009

are generally weak and, in fact, opposite in sign to the relationships for the SeaWiFS period. For 1998-2007, the minimum barotropic streamfunction experienced a positive anomaly, mixed layers shoaled, and horizontal fluxes declined in the NE region. For 1949-2009, positive anomalies of the minimum barotropic streamfunction are, instead, weakly associated with increased horizontal nutrient fluxes ($R_{NE(PsiMin,Horiz)}$ = 0.25). At the same time, shallower mixed layers (a negative anomaly) are associated with decreased vertical

fluxes and increased horizontal fluxes ($R_{NE(MLD,Vert)}$ = 0.52, $R_{NE(MLD,Horiz)}$ = -0.42). In the NW region, light limitation is clearly the driver of biomass changes on both timescales (Fig. 1,4, Table 1), but large negative

anomalies of vertical fluxes and positive anomalies of horizontal fluxes also occur as mixed layers shoal over 1997-2008 (Fig. 9, 10). However, for 1949-2009 in the NW, the reverse is found; vertical flux convergence increases and horizontal nutrient flux convergence decreases coincident with shallower mixed layers ($R_{NW(MLD,Vert)}$ = -0.33, $R_{NW(MLD,Horiz)}$ = 0.46). Long-term correlations of physical changes to nutrient fluxes in the two subpolar regions differ from those occurring with 1998-2007 trends is consistent with the weak long-term correlations that explain no more than 30% of the variance. The lack of consistent associations between biomass and physical variability over both timescales illustrates the complexity of the system and makes clear that relationships revealed by relatively short-lived observing systems are not necessarily representative of the long term.

**4 Discussion**

The decline in the strength of the subpolar gyre modeled here (Fig. 6) is consistent with observations in the North Atlantic since the mid-1990s [Häkkinen and Rhines, 2004; Hátún et al., 2005; Våge et al., 2008; Foukal and Lozier, 2017]. We show here that these physical changes have the potential to drive substantial impacts on the light field and on vertical and horizontal nutrient supply. These changes are sufficient to explain the modeled biomass trends over 1998-2007 that are, in turn, consistent with satellite observations (Fig 1, S1).

Foukal and Lozier [2017] provide an updated analysis with respect to the relationship of physical changes in the gyre to the East Atlantic Pattern (EA) and the North Atlantic Oscillation (NAO). While the EA indicates the position of the westerly winds, NAO indicates their strength [Foukal and Lozier, 2017; Comas-Bru and McDermott, 2014]. A preliminary investigation using the winter (DJFM) EA index from NOAA CPC indicates that only in the nutrient-limited SE region are there significant correlations. Biomass is correlated to the EA ($R_{SE(EA,Biomass)}$ = 0.48), a relationship apparently driven by horizontal flux convergence ($R_{SE(EA,Horiz)}$ = 0.43). In the SE region, biomass is not significantly correlated to the winter (DJFM) NAO [Hurrell and NCAR, 2017], which may be due to significant opposing impacts of the NAO on horizontal and vertical nutrient flux convergence ($R_{SE(NAO, Horiz)}$ = 0.37, $R_{SE(NAO, Vert)}$ = -0.30). These correlations are all zero-lag. We do not find stronger correlations when biomass lags the EA or NAO by up to 3 years. That there are no significant correlations north of 50 $^o$N, in the NE and NW regions, between these climate modes and biomass is consistent with the weak correlations of horizontal and vertical flux convergence to physical fields (Table 2).

Williams and Follows [1998] illustrate that on the mean, horizontal Ekman fluxes in the surface are critical to nutrient supply in the North Atlantic from 40-60 $^o$N, particularly for the northeast subtropical gyre. Yet, Williams et al. [2000] find Ekman nitrate flux variability to be an order of magnitude smaller than convective flux variability in this region. We find that 0-100 m horizontal nutrient convergence contributes 25-35% of the mean nutrient supply in our two regions to the east of 35 $^o$W. In contrast to Williams et al. [2000], we do find horizontal flux convergence to be important to variability, with the 1949-2009 standard deviation of horizontal flux convergence in the SE region being 66% of the standard deviation of the sum of vertical and

horizontal, while the vertical flux convergence standard deviation is 95% of the sum. In the NE region, vertical and horizontal flux convergence are anti-correlated ($R_{NE(Vert,Horiz)}$ = -0.64, Table 2) such that their variability partially cancels. The standard deviation of vertical flux convergence here is 125% of the sum, while the standard deviation of horizontal flux convergence is 108% of the sum. These very different findings can at least be partially attributed to the fact that Williams et al. [2000] had only a climatological nitrate data field to couple to their mixed layer model and windstress-based Ekman divergence calculation. The use of a smooth climatological nutrient field would not likely allow for the strong co-variance between vertical and horizontal supply terms that these run-time diagnostics are able to reveal. As variability of nutrient supply to the surface ocean is critical to subpolar North Atlantic biomass variability, datasets that temporally resolve upper ocean nutrient fields would be most valuable to future studies. Large-scale deployment of autonomous floats with biogeochemical sensors will be essential to the development of these critical datasets [Johnson et al. 2009].

In our model, reduced horizontal nutrient supply over the SeaWiFS period (1998-2007) drove the observed reductions in biomass on the northeastern flank of the North Atlantic subtropical gyre (our SE region). This mechanism contrasts to previous analyses that attribute the observed changes to locally increased stratification and the associated reduced vertical supply of nutrients [Behrenfeld et al. 2006; Polovina et al. 2008; Martinez et al. 2009] or to subtle shifts in the balance between phytoplankton accumulation and loss [Behrenfeld 2014]. Contrasting mechanisms in two different timeframes are found in this region. For 1949-2009, simulated biomass and SST are anti-correlated ($R_{SE(Biomass, SST)}$ = -0.57, Table 2) while biomass anomalies are positively correlated to both vertical and horizontal nutrient supply changes ($R_{SE(Biomass,Vert)}$ = 0.63, $R_{SE(Biomass,Horiz)}$ = 0.44). However, over the SeaWiFS period, reduced horizontal flux convergence is more important than changes in vertical flux convergence to simulated biomass declines (Fig. 10). The fact that horizontal processes are important on both timescales is consistent with previous findings that horizontal nutrient fluxes are seasonally important [Dave et al. 2015] and also that SST as a proxy for stratification is alone insufficient to describe biomass interannual variability in this region [Lozier et al., 2011]. It is reasonable to expect similar mechanisms operate on the edges of the subtropical gyres elsewhere around the globe. Particularly in these intergyre regions, three-dimensional perspective on nutrient supply should be taken when observations are interpreted and when expected mechanisms of future change are considered [Doney 2006; Bopp et al. 2013].

In the context of 21[st] century climate-driven changes in biomass, Laufkötter et al. (2015) find zooplankton grazing to be important to biomass in some models under a strong climate change forcing scenario (RCP8.5). Zooplankton is not the driver of biomass changes in this model (Figure S3), with the very different timescales and levels of forcing for change -- 10 years of interannual variability in this study, ~100 years with strong forcing in Laufkötter et al. (2015) -- likely being a factor in this difference. That zooplankton grazing is not temperature dependent in this model may also contribute, but any potential effects would be limited by the annual mean temperature change from 1998-2000 to 2005-2007 being substantially smaller (+0.02 $^o$C, +0.28

$^{o}$C, +0.13 $^{o}$C, in SE, NE and NW regions, respectively) than over the 21$^{st}$ century in the RCP8.5 scenario (1-4 $^{o}$C at 40-60 $^{o}$N, Laufkötter et al. 2015).

In this simulation, North Atlantic biomass variability to the north of 40 $^{o}$N is quite heterogeneous and dependent on different mechanisms at distinct locations, with the dominant mechanisms shifting across timescales. Though a large-scale averaging approach may be appropriate for some biogeochemical studies [Fay and McKinley, 2013, 2014], relationships between the surface ocean carbon cycle and productivity may not be well captured by correlations over large-scale ocean biomes that take the whole of the subpolar gyre
as one region [Fay and McKinley, 2017]. An approach using smaller subregions will likely support a deeper understanding of biological coupling to the carbon cycle in this region.

These findings suggest myriad directions for further analysis. In order to address the simplest measure of change, we use annual mean fields for both the observations and the model. A deeper consideration of how these changes operate in the context of the significant seasonality of the region would be very interesting.
This analysis does not elucidate how variability in physical supply of silicate impacts biomass variability. Particularly considering long-term correlations between physical drivers and phosphate supply in NE and NW region that are opposite to those evidenced for 1998-2007, a complementary analysis of variability in silicate supply may be useful. Similarly, it would be of value to study the relative impacts of large and small phytoplankton size classes on total biomass variability particularly in the northwest region where light and
nutrient limitation drive biomass in opposite directions (Table 1). Assessment of the modulation of subsurface nutrient fields by subpolar gyre physical changes, and in turn how these subsurface changes influence surface biomass would be worthwhile. Spatial analysis based on empirical orthogonal functions [Breeden and McKinley, 2016] could illustrate the dominant large-scale modes of biomass variability and may reveal the degree to which climate modes impact biomass on longer timescales. With respect to the
period of satellite observations, a numerical simulation that covers both the SeaWiFS and MODIS period would allow study of the period since 2007 in which 1998-2007 trends appear to largely have reversed (Fig. 2). For such a simulation, greater physical resolution should improve representation of the gyre structure and its variability. Though the current ecosystem is able to capture the large-scale patterns of biomass change remarkably well, it would also be valuable to assess the impact of different levels of ecosystem complexity
in future modeling work.

**5 Conclusions**

In the North Atlantic from 40-60 $^{o}$N over 1998-2007, biomass estimated from SeaWiFS ocean color increases to the west of 30-35 $^{o}$W and declines to the east. A regional coupled physical-biogeochemical model that reproduces 1998-2007 trends indicates that the relief of light limitation with shoaling mixed layers was
sufficient to drive the observed biomass increase in the west. This model attributes biomass declines to the east of 30-35 $^{o}$W over 1998-2007 to reduced nutrient supply. On the northeastern flank of subtropical gyre, in our southeast region, changing horizontal nutrient supply drives biomass change over the SeaWiFS period.

For the full model experiment, 1949-2009, both horizontal and vertical nutrient supply are important to interannual variability here. In the northeast subpolar gyre, horizontal nutrient supply is the most important driver of biomass variability both for the 1998-2007 SeaWiFS period and for the full model experiment.

Though nutrient supply in three dimensions is can explain biomass changes to the east of 30-35 $^{\circ}$W, clear connections between these supply terms and large-scale physics or climate indices are elusive. Neither does the minimum barotropic streamfunction or local mixed layer depths consistently explain nutrient flux variability both for the satellite period and the full model experiment. In the southeast, biomass variability over 1949-2009 weakly correlates to the East Atlantic (EA) pattern, but nowhere is the NAO correlated with biomass variability. Given this evidence that horizontal and vertical nutrient supply are important to observed biomass variability, and evidence from other studies that these modes of climate influence the gyre strength, currents, deep mixing, and Ekman suction and divergence of the subpolar gyre, more investigation of the links between North Atlantic climate and biomass variability is clearly warranted.

**Acknowledgments**

The authors thank Val Bennington, Lucas Gloege, Dierk Polzin, Melissa Breeden, and Amanda Fay for their assistance with the model and datasets; and the Oregon State Ocean Productivity group for providing satellite-based productivity and biomass estimates. We thank two anonymous reviewers and Scott Doney for their careful reviews. We gratefully acknowledge funding from NASA (NNX/11AF53G, NNX/13AC53G, NNX13AC94G).

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

**Tables**

**Table 1: Correlations of phosphate and light limitation for small phytoplankton to large and small biomass, 1949-2009.** For conciseness, shown here are only small phytoplankton limitation terms, but since these are ratios calculated from identical fields, the correlations are very similar large phytoplankton (Table S1). Bold indicates statistical significance ($p < 0.05$). Detrending is applied prior to correlation analysis.

| **SOUTHEAST** | large biomass | phosphate limitation, small phytoplankton | light limitation, small phytoplankton |
|---|---|---|---|
| small biomass | **0.52** | **0.68** | **-0.75** |
| large biomass | - | **0.74** | **-0.57** |
| phosphate limitation, small | | - | **-0.87** |

| **NORTHEAST** | large biomass | phosphate limitation, small phytoplankton | light limitation, small phytoplankton |
|---|---|---|---|
| small biomass | **0.36** | -0.09 | -0.08 |
| large biomass | - | **0.31** | -0.17 |
| phosphate limitation, small | | - | **-0.87** |

| **NORTHWEST** | large biomass | phosphate limitation, small phytoplankton | light limitation, small phytoplankton |
|---|---|---|---|
| small biomass | **-0.34** | **-0.50** | **0.61** |
| large biomass | - | **0.83** | **-0.63** |
| phosphate limitation, small | | - | **-0.71** |


**Table 2: Correlations of biomass to physical drivers and horizontal and vertical phosphate flux convergence, 1949-2009**. The minimum barotropic streamfunction is found within 60-30 °W, 50-65 °N; maximum mixed layer depth (MLD) and sea surface temperature (SST) are area-weighted averages for each of the three averaging regions. Bold indicates statistical significance ($p < 0.05$). Detrending is applied prior to correlation analysis.

| SOUTHEAST | Minimum Barotropic Streamfunction | Maximum MLD | SST | Horizontal | Vertical |
|---|---|---|---|---|---|
| Biomass | **-0.37** | **0.54** | **-0.57** | **0.44** | **0.63** |
| Minimum Barotropic Streamfunction | - | **-0.42** | **0.38** | -0.18 | -0.23 |
| Maximum MLD | | - | **-0.65** | 0. 10 | **0.66** |
| SST | | | - | -0.02 | **-0.64** |
| Horizontal | | | | - | **-0.29** |

| NORTHEAST | Minimum Barotropic Streamfunction | Maximum MLD | SST | Horizontal | Vertical |
|---|---|---|---|---|---|
| Biomass | 0.18 | -0.02 | 0.08 | **0.48** | -0.25 |
| Minimum Barotropic Streamfunction | - | **-0.60** | **0.72** | **0.25** | -0.22 |
| Maximum MLD | | - | **-0.76** | **-0.42** | **0.52** |
| SST | | | - | **0.34** | **-0.28** |
| Horizontal | | | | - | **-0.64** |

| NORTHWEST | Minimum Barotropic Streamfunction | Maximum MLD | SST | Horizontal | Vertical |
|---|---|---|---|---|---|
| Biomass | 0.14 | **0.28** | -0.23 | **0.69** | **-0.42** |
| Minimum Barotropic Streamfunction | - | **-0.42** | **0.67** | 0.08 | -0.10 |
| Maximum MLD | | - | **-0.42** | **0.46** | **-0.33** |
| SST | | | - | **-0.52** | **0.43** |
| Horizontal | | | | - | **-0.76** |

## Figures

**Figure 1: Surface ocean biomass** (a) estimated from SeaWiFS using the CbPM model, (b) 0-100m modeled biomass, (c) SeaWiFS trend 1998-2007, and (d) 0-100m modeled biomass trend 1998-2007. In c and d, significant trends are marked with a black contour. In d, the three focus regions are outlined in red.

**Figure 2: Annual anomalies of surface ocean biomass** for SeaWiFS (1998-2007, red), MODIS (2003-2015, blue) and model (1998-2009, 0-100m, black) (a) SE region, (b) NE region, and (c) NW region. The corresponding monthly timeseries is shown in Fig. S1.

**Figure 3: Phosphate** (a) World Ocean Atlas (Garcia et al. 2006) (b) 0-100m modeled phosphate, (c) 0-100m modeled phosphate tred 1998-2007. In c, significant trends are marked with a black contour and the three focus regions are outlined in red.

**Figure 4: Surface ocean small and large phytoplankton biomass** (a) 0-100m modeled small phytoplankton biomass, (b) 0-100m modeled large phytoplankton biomass, (c) small phytoplankton trend 1998-2007, and (d) large phytoplankton trend 1998-2007. In c and d, significant trends are marked with a black contour and the three focus regions are outlined in red.

**Figure 5: Terms for limitation** by (a) phosphate, large phytoplankton (Eq. 3) (b) Light, small phytoplankton (Eq. 2), (c) 1998-2007 trend in phosphate limitation, and (d) 1998-2007 trend in light limitation. All are unitless. In c and d, significant trends are marked with a black contour and the three focus regions are outlined in red.

**Figure 6: Barotropic streamfunction** (a) 1998-2000 mean and (b) 2005-2007 mean. The zero streamfunction contours between 55-15$^{\circ}$W for each period (bold black) and for the 1998-2007 mean (white) are marked. See Figure S5 for map of 1998-2007 trend.

**Figure 7: Maximum mixed layer depths** for (a) 1998-2000 mean and (b) 2005-2007 mean. Timeseries of monthly mixed layers for (c) SE region, (d) NE region, and (e) NW region.

**Figure 8: Phosphate diagnostics**, 0-100m flux convergence (mmol m$^{-3}$ yr$^{-1}$) for (a) vertical, (b) horizontal, (c) net physical, and (d) biological. Biological is negative because biomass removes phosphate from the surface ocean. The three focus regions are outlined in red in each panel.

**Figure 9: Phosphate diagnostics 1998-2007 annual timeseries**, 0-100m flux convergence (mmol m$^{-3}$ yr$^{-1}$) for (a) SE region, (b) NE region, and (c) NW region.

**Figure 10: Phosphate diagnostics 1998-2007 trends,** 0-100m flux convergence trend (mmol m$^{-3}$ yr$^{-2}$) for (a) vertical, (b) horizontal, (c) net physical, and (d) biological. Positive biological trends are consistent with negative biomass trends because less phosphate is removed as less biomass is formed. Significant trends are marked with a black contour and the three focus regions are outlined in red in each panel.

Figure 1

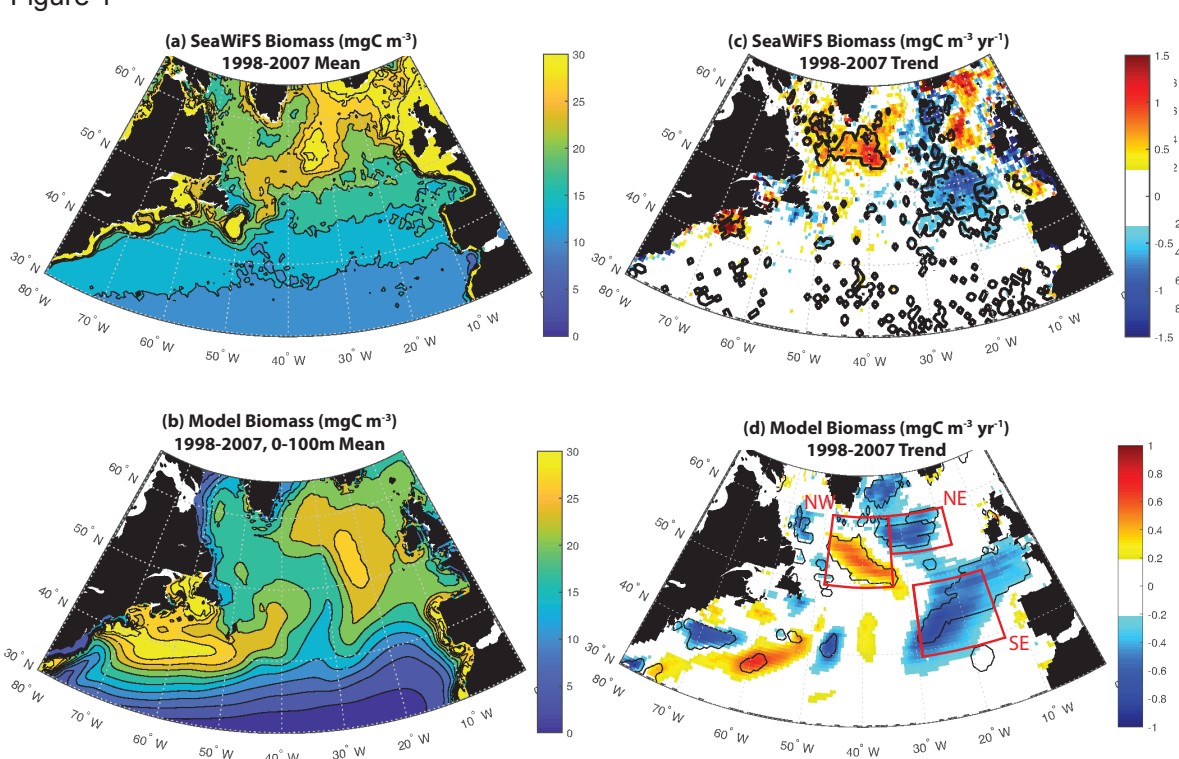

Figure 2

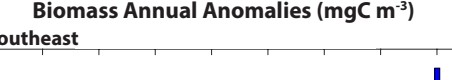

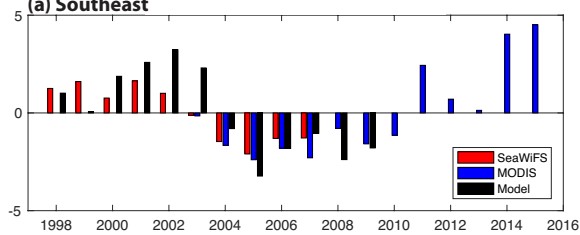

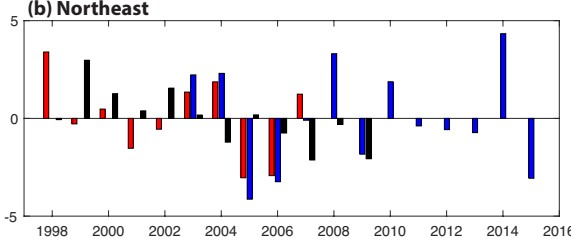

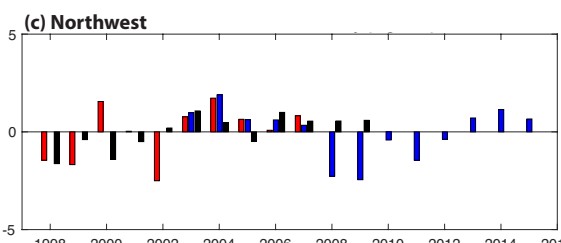

Figure 3

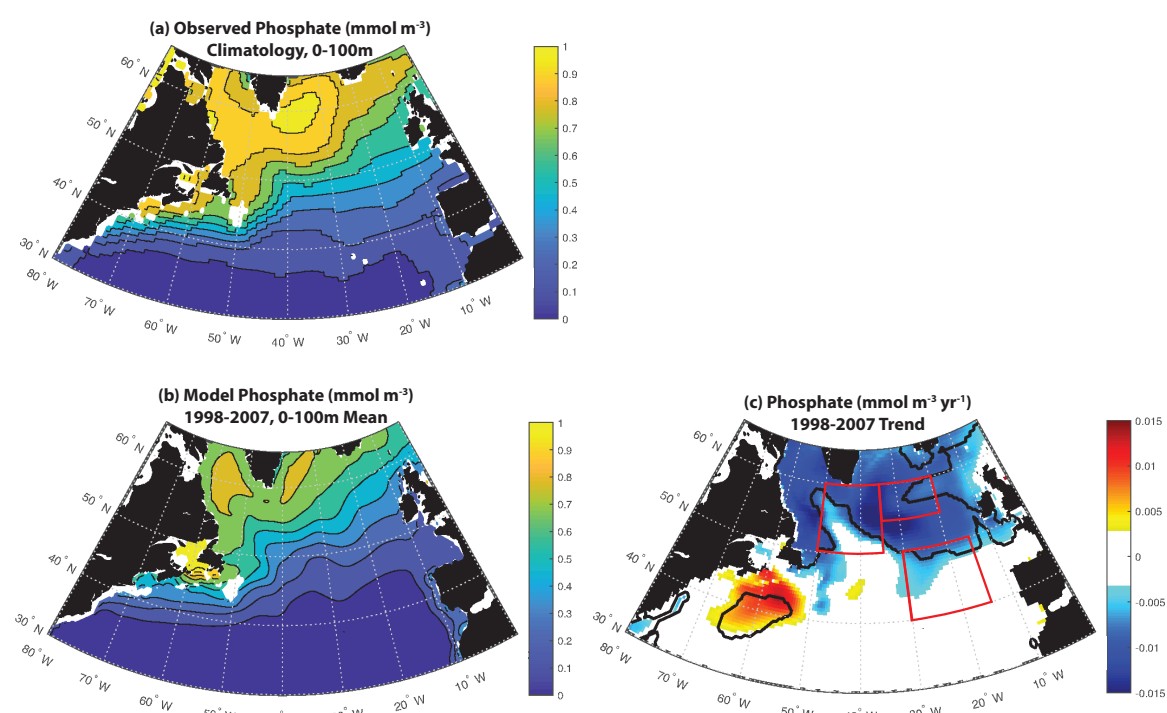

Figure 4

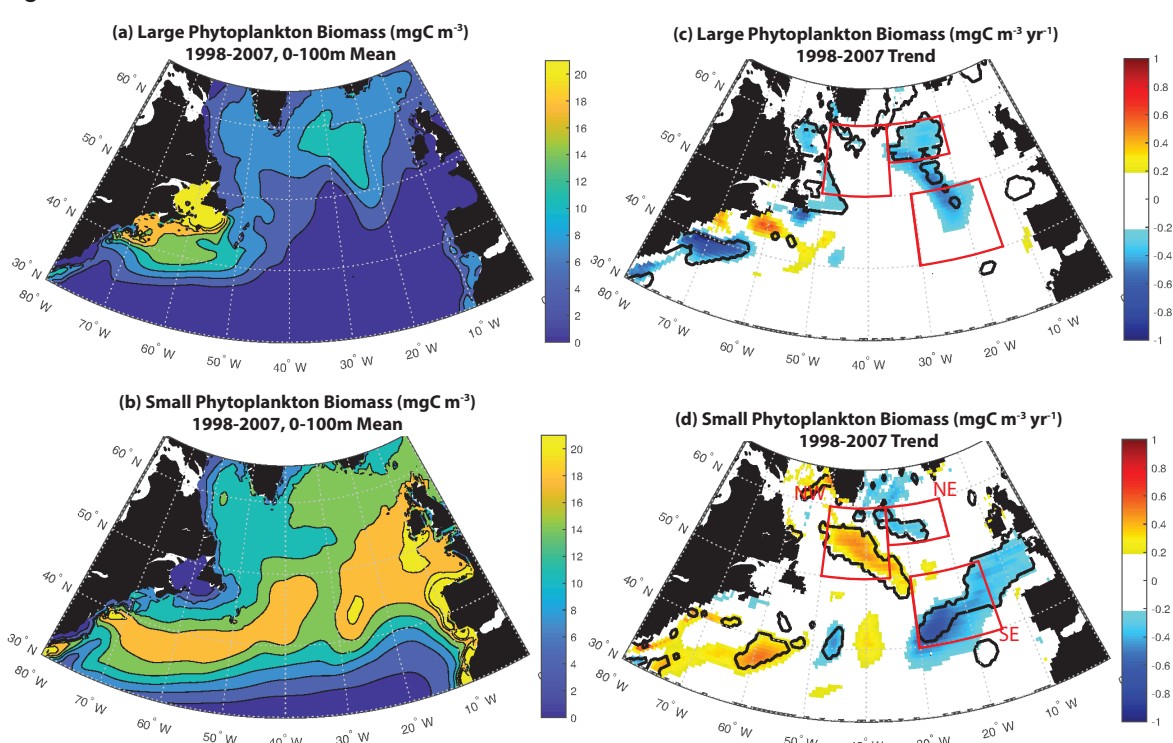

Figure 5

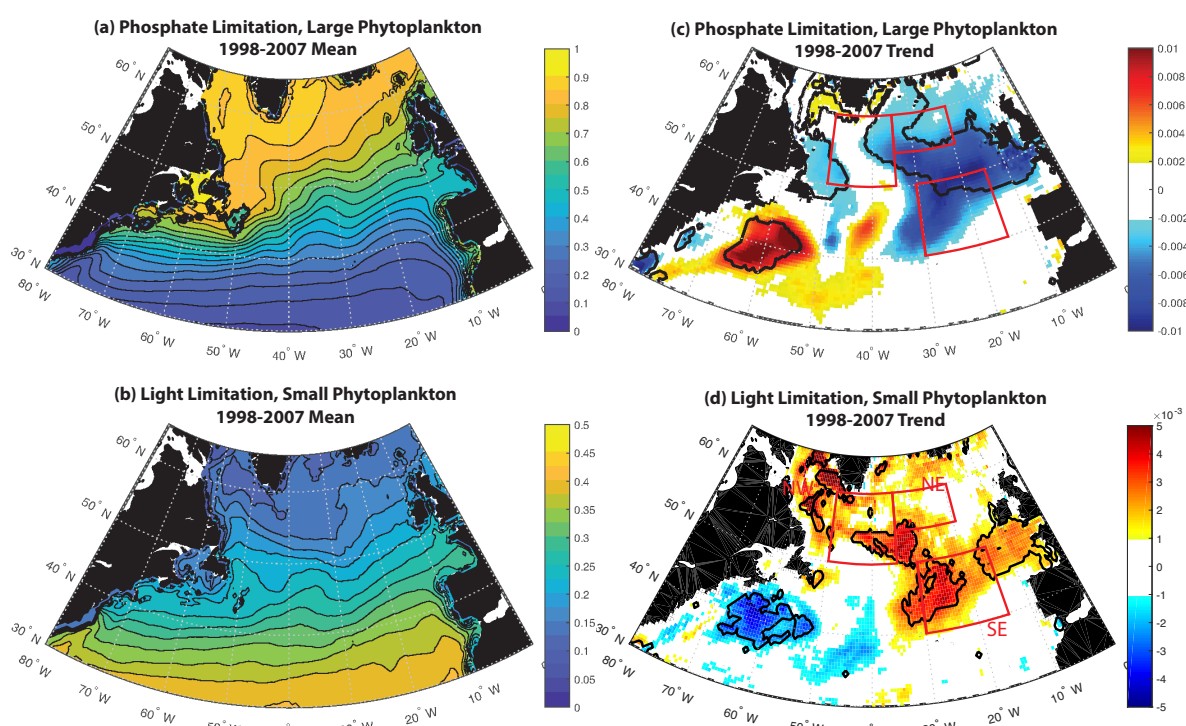

Figure 6

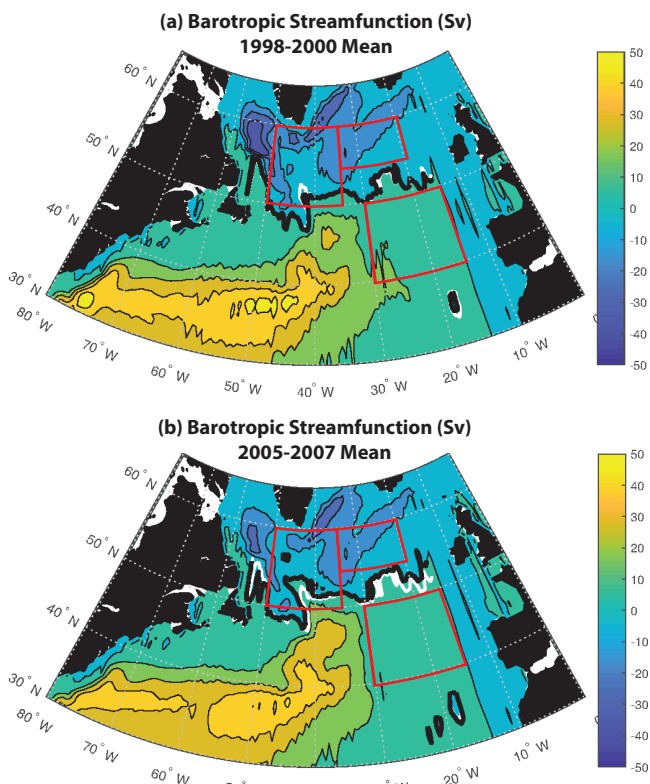

Figure 7

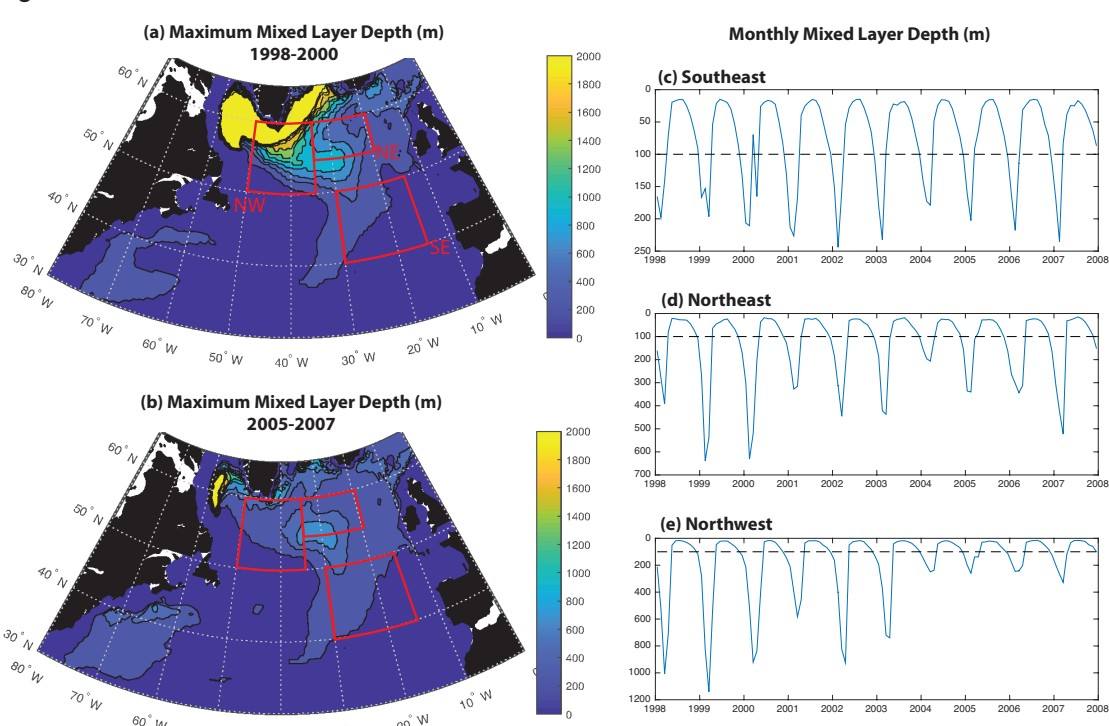

Figure 8

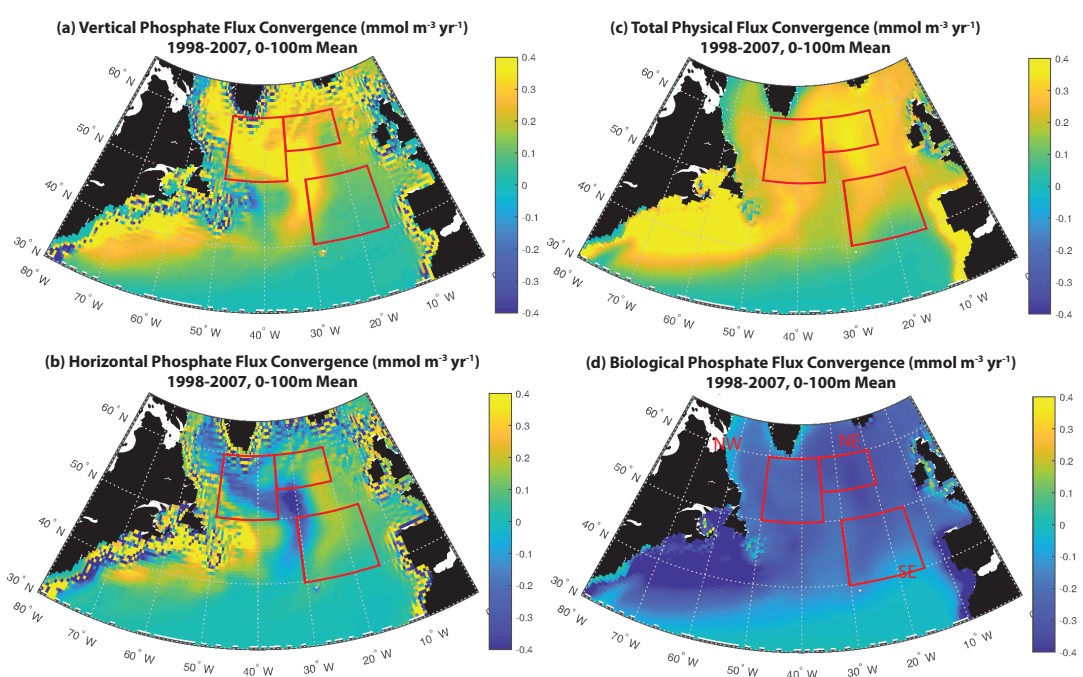

Figure 9

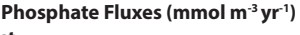

**Phosphate Fluxes (mmol m⁻³ yr⁻¹)**

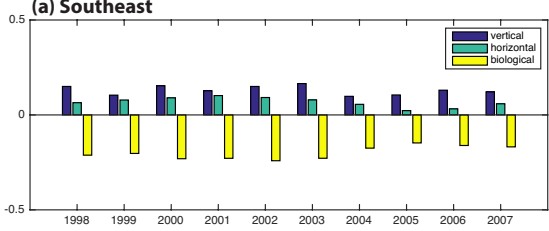

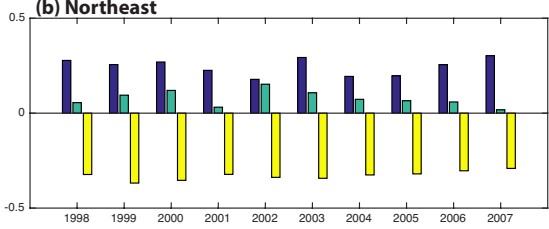

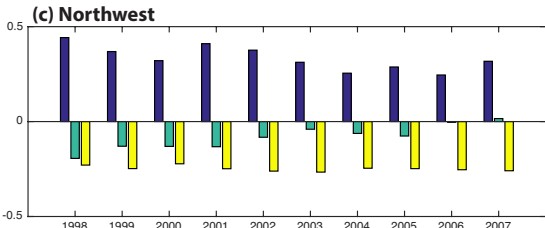

Figure 10

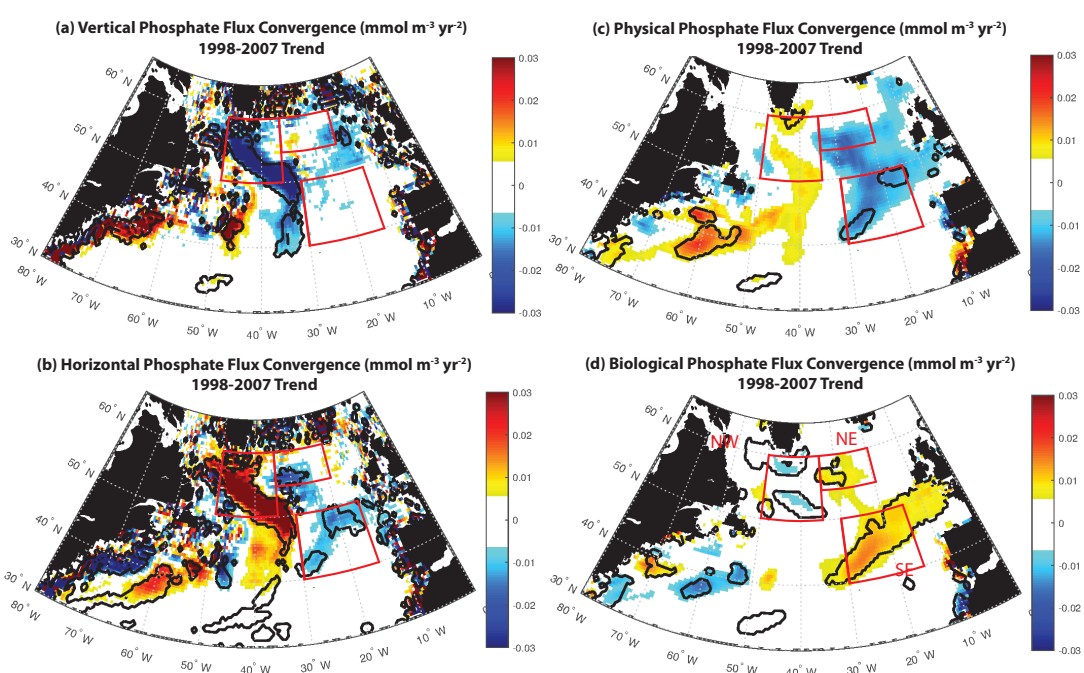

**(a) Vertical Phosphate Flux Convergence (mmol m⁻³ yr⁻²)**
**1998-2007 Trend**

**(c) Physical Phosphate Flux Convergence (mmol m⁻³ yr⁻²)**
**1998-2007 Trend**

**(b) Horizontal Phosphate Flux Convergence (mmol m⁻³ yr⁻²)**
**1998-2007 Trend**

**(d) Biological Phosphate Flux Convergence (mmol m⁻³ yr⁻²)**
**1998-2007 Trend**