# Peer review of "Mechanisms of northern North Atlantic biomass variability"

_Biogeosciences, 2018_

## Referee Comment (RC1) · Anonymous Referee #1 · 6 Apr 2018

This manuscript explores some of the mechanisms controlling phytoplankton biomass variability in the North Atlantic over the later 20th century and early 2000s. The manuscript is interesting and well written, and to some extent appears to challenge the view that nutrient dependent biomass variability is controlled only by the vertical nutrient supply. However, I have some reservations with the method adopted and cannot therefore recommend immediate publication. My main issue with the manuscript is the approach of using correlation coefficients between biomass and light/phosphate limitation terms as a means of attributing causality. This seems to be something of a shortcut given that these factors are likely to be somewhat collinear with other potential drivers of phytoplankton biomass. If possible, I think a more complete approach would be to recompute the model phytoplankton biomass using the limitation terms and other

drivers (similar to what is done in Laufkötter et al., 2015 for several biogeochemistry models). This would allow the authors to assess the separate impacts of bottom-up processes (the influence of limitation terms on growth rates) as well as top-down loss terms (mortality/grazing). If this approach is not possible due to a lack of model output then I think some of the paper's conclusions should be toned down especially when using these correlation coefficients to infer the processes driving SeaWiFS variability.

My other issue is that a number of processes that could be responsible for some of the trends in biomass variability seem to be neglected. These are perhaps not included in the model but this should nonetheless be stated. What role does temperature play? Is umax independent of temperature? What about zooplankton grazing rates? If grazing is temperature dependent does this explain any of the biomass variability? These sorts of things may be important given that certain models seem to show phytoplankton biomass declines despite increases in phytoplankton growth rates, due to overwhelming increases in losses to zooplankton grazing (Laufkötter et al., 2015).

Specific comments

Ln 90. What was the decision behind the use of the CbPM algorithm? Given that alternative algorithms can substantially differ it would be good to know that the trends described are robust to this algorithm choice. Perhaps a supplementary figure could be produced comparing CbPM mean state and trends in this region with an alternative algorithm such as VGPM.

Ln 107. I think more model details are needed here even if they are published elsewhere. Specifically, what is meant by a "phosphorus-based ecosystem"? It would be good to have some mention of N. Is everything assumed to be Redfield? If so, is this a potential limitation of using this sort of model in this context? Is there any N fixation in the model?

Ln 140-160. See general comments above. Where does temperature limitation fit in? If not at all then I think this should be mentioned. Also, this section focuses on the

effects of limitation terms on growth rates yet the analysis focuses on biomass not growth rates. I think the authors could better describe how growth rates and biomass are related, mentioning the additional processes that affect biomass in their model (e.g. zooplankton grazing?).

Ln 401-404. Although declines in the horizontal nutrient supply may be the proximate driver is the ultimate driver not declines in the vertical supply to the west of the SE box? If so, perhaps this statement should be more nuanced.

Figs 6 and 7. The differences between panels a and b are difficult to see in these plots. It would be useful to add a panel to each of these figures that is the difference between these time slices.

Technical/minor corrections

Ln 25-28. I think some references are needed in this paragraph.

Ln 38. Type error. "...do not fit their..."

Ln 44-48. Within this context it might be useful to mention that Kwiatkowski et al., 2017 related interannual variability of productivity to long-term trends across an ESM ensemble. Capturing productivity variability may therefore help reduce long-term projection uncertainties.

Ln 74-75. "substantial change" reads as if there has been a climatic shift in the North Atlantic subpolar gyre. I think the authors are only referring to variability here and should clarify this.

Ln 101. Space before units for consistency. "2200 m"

Ln 109. I think something should come after "small". Small phytoplankton or nanophytoplankton?

Ln 172. Space before units for consistency. "100 m"

Ln 224. I don't think "all three timeseries" is correct looking at Fig 2a. MODIS does not appear to have any positive anomalies prior to 2004.

Ln 246. Looking at Fig 4 small phytoplankton appear to dominate in the North (> 52°N). Although to a lesser extent than in the South. If correct, this sentence should be amended.

Ln 269. To say "only 40% of total biomass" seems strange.

Ln 309. I would change the word "collaborative".

Ln 361-364. This seems more suitable to the discussion than the results.

Ln 367. Type error. Remove "in" or "since".

Ln 395. Type error. The use of "a" smooth climatological nutrient. . .

Ln 409. Type error. . . ."on" the edges of. . .

Ln 414-415. Suggest improving sentence readability. Perhaps: . . ."with the dominant mechanisms shifting across timescales"

Ln 418. It is not clear to me what is meant by a "granular approach".

Ln 419. Type error. . . ."in" this region.

Ln 425. Type error. . . ."in the" northwest region.

Ln 434. Type error. "of" value or "valuable"

Ln 443. Type error. Remove "in".

References

Laufkötter, C., Vogt, M., Gruber, N., Aita-Noguchi, M., Aumont, O., Bopp, L., et al. (2015). Drivers and uncertainties of future global marine primary production in marine ecosystem models. Biogeosciences, 12(23), 6955–6984. https://doi.org/10.5194/bg-12-6955-2015

Kwiatkowski, L., Bopp, L., Aumont, O., Ciais, P., Cox, P. M., Laufkötter, C., et al. (2017). Emergent constraints on projections of declining primary production in the tropical oceans. Nature Climate Change. https://doi.org/10.1038/nclimate3265

---

## Referee Comment (RC2) · Anonymous Referee #2 · 10 Apr 2018

Review of "Mechanisms of northern North Atlantic biomass variability" Galen A. McKinley, Alexis L. Ritzer, and Nicole S. Lovenduski Biogeosciences, submitted.

Summary The manuscript integrates satellite ocean color observations and a coupled ocean physical-biogeochemical model to quantify plankton trends in the temperate and subpolar North Atlantic and evaluate potential underlying mechanisms. The model is an essential component for identifying physical transport effects on plankton dynamics. The study builds on a substantial literature on this important scientific question.

Overall, I found the manuscript to be relatively weak. The modeling study focuses on linear trends that are likely not robust over such a small time window (1998-2007) for SeaWiFS data. The trend analysis leaves out the longer integrated SeaWiFS-MODIS data set, which exhibits substantial interannual to decadal variability, often with different

temporal patterns than inferred from the shorter linear trend analysis. The numerical modelling is also limited to 2009 and thus is not compared with later MODIS data.

ocean and is a significant contribution that fits well within the scope of Biogeosciences journal. The manuscript is generally well written.

Methodology The manuscript utilizes well-documented satellite ocean color data from SeaWiFS and MODIS-Aqua and an established ocean physical-biogeochemical model and hindcasting techniques. The study utilizes a set of monthly diagnostics for the physical and biological terms affecting the phosphate budget. The biological diagnostics are available for the SeaWiFS period (1998-onward) but not prior to 1998 (model output lost; Line 135). This subtracts some from the trend analysis over longer time periods 1949-2009 where only model biomass is available. Also, it is not clear why the model hindcast stops in 2009, now more than 8 years ago.

Results One limitation with the analysis is the focus on linear trends over a relatively short analysis window (1998-2007) (Figure 1). As is clearly shown by Figure 2, the regional temporal patterns primarily exhibit inter-annual to decadal trends and any linear trend is relatively small and sensitive to the choice of time window (an issue that the lead author is well familiar with and has published on previously). For example, the manuscript identifies declining biomass east of 30-35 deg. W (e.g. Line 11-12; Line 201-202), however, this is not consistent with the observations. The SeaWiFS-era trends in (Figure 1c) show only a small region of declining trends in the northeast (north of 55 deg.) with a substantial region of positive trends (though admittedly not statistically significant). The regional trends in Figure 2 that include MODIS data do not show such clear trends, and in fact from the merged SeaWiFS-MODIS data the trend in the southeast actually change signs.

The manuscript would be much stronger if the focus was expanded to include the agreement (or disagreement) of model and observed interannual variability and underlying mechanisms. At a minimum, there needs to be more up front discussion of the

rationale for and limitations of focusing on linear trends.

The singular focus on annual means in the model and data analysis neglects the substantial seasonality in bloom dynamics in the subpolar North Atlantic. This raises two issues. First, there is no discussion of the robustness of annual mean satellite observations because of sampling biases, particularly during winter. Second, it is not clear if annual mean biomass is the biologically most important indicator; would more relevant indicators be peak surface biomass concentration or peak integrated biomass (ala arguments of Berhenfeld and Boss). Implicitly the analysis also assumes that only bottom-up factors (light and nutrient limitation) influence trends in phytoplankton biomass, neglecting possible top-down factors. This may be true for the model, but perhaps is an incomplete picture of the actual ocean.

Specific comments Line 180: I have some concerns regarding the following paragraph: "This analysis is based on annual mean fields. A 3-month lag of the biology diagnostics and biomass fields after physical diagnostics and other physical fields is employed to account for the maximum physical forcing occurring in the winter prior to the spring bloom. Thus, annual mean physical fields are averaged from October of the prior year to September of the year in question. Biological fields are January to December averages." I understand the need perhaps to adjust the year window to capture the relevant Fall and early winter pre-conditioning of subsequent spring bloom, but the text is not framed in terms of pre-conditioning. Rather a somewhat arbitrary 3-month lag is argued, inconsistent with the well-observed latitudinal seasonal patterns in the timing or phenology of bloom dynamics for the North Atlantic. Further, it is not clear that the relevant physical quantities are annual means for variable such as mixed layer depth with large seasonal variation and where it is more likely that the maximum winter mixed layer depth is more biologically relevant. Given the richness of monthly model output, a more nuanced data analysis would be warranted.

Line 209-212 "In both observations and models, the magnitudes of these changes are large in comparison to the mean. In the declining regions, where mean biomass is 15-

25 mgC m-3 (Fig. 1, S1), trends -0.5 to -1.5 mgC m-3 yr-1 over 10 years imply biomass reductions of 30-50 %. To the west of 30-35 deg. W, increases of a comparable percentage are implied." The magnitude of these trends may be appropriate for some pixel level trends but the magnitude of the trends are roughly an order of magnitude smaller at the regional scale in Figure 2.

Figures 1 and 2; Lines 213-219: Regional analysis boxes are identified for the model in Figure 1d and linked to the time-series in Figure 2. It appears from the text that the same regional boxes are used for model and satellite observations, but it is unclear if this is appropriate given the spatial mismatch in the model and observed mean and trend patterns. The text (Lines 226-229) argue that this has a minimal effect but this should probably be shown in some figures in the supplement.

Line 237-239: There is considerable nutrient data (though still sparse) from the CLIVAR Repeat Hydrography for the sub-polar North Atlantic, particularly from the German and Canadian occupied lines; worth looking in to.

Line 250: Are the annual means for phosphate and light limitation terms simply the straight means? Was there any consideration of weighting the limitation with seasonal variations in biomass or NPP, which might enhance the biological relevance.

Line 285: Does the model mixed layer trend agree with observations?

Line 296-299: The wording of the text here is awkward; would be phrased as, for example, "horizontal advective divergence" (or convergence), etc. As written, the text and figure labels (Figure 8 and 9) confuse "flux" with "flux divergence".

---

## Author Comment (AC1) · 9 May 2018

RESPONSE to Anonymous Referee #1 This manuscript explores some of the mechanisms controlling phytoplankton biomass variability in the North Atlantic over the later 20th century and early 2000s. The manuscript is interesting and well written, and to some extent appears to challenge the view that nutrient dependent biomass variability is controlled only by the vertical nutrient supply. However, I have some reservations with the method adopted and cannot therefore recommend immediate publication. My main issue with the manuscript is the approach of using correlation coefficients between biomass and light/phosphate limitation terms as a means of attributing causality. This seems to be something of a shortcut given that these factors are likely to be somewhat collinear with other potential drivers of phytoplankton biomass. If possible, I think

a more complete approach would be to recompute the model phytoplankton biomass using the limitation terms and other drivers (similar to what is done in Laufkötter et al., 2015 for several biogeochemistry models). This would allow the authors to assess the separate impacts of bottom-up processes (the influence of limitation terms on growth rates) as well as top-down loss terms (mortality/grazing). If this approach is not possible due to a lack of model output then I think some of the paper's conclusions should be toned down especially when using these correlation coefficients to infer the processes driving SeaWiFS variability.

We address this concern by plotting zooplankton trends in the model over the main analysis period (new Figure S2 and included here). This figure shows that zooplankton trends occur of the same sign as of biomass. Were top-down processes driving the declines (increases) in biomass, then one would expect to see increasing (decreasing) zooplankton trends in the southeast and northeast (northwest). This is not what is occurring in the model. It is clear that nutrient and light trends are responsible for the modeled trends. We include mention of this analysis in the main text, with reference to this new Figure S2.

My other issue is that a number of processes that could be responsible for some of the trends in biomass variability seem to be neglected. These are perhaps not included in the model but this should nonetheless be stated. What role does temperature play? Is umax independent of temperature? What about zooplankton grazing rates? If grazing is temperature dependent does this explain any of the biomass variability? These sorts of things may be important given that certain models seem to show phytoplankton biomass declines despite increases in phytoplankton growth rates, due to overwhelming increases in losses to zooplankton grazing (Laufkötter et al., 2015).

Thank you. We add mention of the temperature dependence of growth in equation 1. Grazing is not temperature dependent and this is also now mentioned. In this model, bottom-up drivers from nutrient and light limitation are responsible for the trends. In the context of zooplankton grazing, we add mention of Laufkötter et al., (2015), also noting

that the difference of findings may be related also to the very different timeframes for trends in that paper (∼100 years), as opposed to this analysis (10 years).

Specific comments Ln 90. What was the decision behind the use of the CbPM algorithm? Given that alternative algorithms can substantially differ it would be good to know that the trends described are robust to this algorithm choice. Perhaps a supplementary figure could be produced comparing CbPM mean state and trends in this region with an alternative algorithm such as VGPM.

Thank you. CbPM is the only algorithm of which we are aware the estimates biomass from satellite. Biomass is the best point of comparison to the model since it is directly carried in the model.

Ln 107. I think more model details are needed here even if they are published elsewhere. Specifically, what is meant by a "phosphorus-based ecosystem"? It would be good to have some mention of N. Is everything assumed to be Redfield? If so, is this a potential limitation of using this sort of model in this context? Is there any N fixation in the model?

Thank you. We clarify that the primary macronutrient in the model is phosphorus, and that silicate is also limiting to the large phytoplankton class. Iron is a micronutrient. Consistent with other lower-complexity ecosystem models, such as BLING (Gailbraith et al. 2010, Biogeoscience), there is no nitrogen in this model.

Ln 140-160. See general comments above. Where does temperature limitation fit in? If not at all then I think this should be mentioned. Also, this section focuses on the effects of limitation terms on growth rates yet the analysis focuses on biomass not growth rates. I think the authors could better describe how growth rates and biomass are related, mentioning the additional processes that affect biomass in their model (e.g. zooplankton grazing?).

Thank you. As shown above, zooplankton grazing follows biomass changes but does

not drive them.

Ln 401-404. Although declines in the horizontal nutrient supply may be the proximate driver is the ultimate driver not declines in the vertical supply to the west of the SE box? If so, perhaps this statement should be more nuanced.

Thank you. We have modified the text to read "to LOCALLY increased stratification" so as to clarify this point.

Figs 6 and 7. The differences between panels a and b are difficult to see in these plots. It would be useful to add a panel to each of these figures that is the difference between these time slices.

We agree that these are somewhat hard to see, but the difference plots are unfortunately not much easier to look at. We have included MLD changes in timeseries form already in Figure 7c-e, and have added the difference plot for barotropic streamfunction in Figure S4.

Technical/minor corrections

Ln 25-28. I think some references are needed in this paragraph. Thank you, references added.

Ln 38. Type error. ". . .do not fit their. . ." Thank you, this has been fixed.

Ln 44-48. Within this context it might be useful to mention that Kwiatkowski et al., 2017 related interannual variability of productivity to long-term trends across an ESM ensemble. Capturing productivity variability may therefore help reduce long-term projection uncertainties. Thank you, we have added this reference.

Ln 74-75. "substantial change" reads as if there has been a climatic shift in the North Atlantic subpolar gyre. I think the authors are only referring to variability here and should clarify this. Thank you. It is stated in the following sentence "There is evidence these changes occur in response to changing buoyancy forcing and wind stress, in

turn associated with modes of climate variability,.." which clarifies the relationship to variability.

Ln 101. Space before units for consistency. "2200 m" Thank you, this has been fixed.

Ln 109. I think something should come after "small". Small phytoplankton or nanophytoplankton? Thank you, this has been fixed.

Ln 172. Space before units for consistency. "100 m" Thank you, this has been fixed.

Ln 224. I don't think "all three timeseries" is correct looking at Fig 2a. MODIS does not appear to have any positive anomalies prior to 2004. Thank you, this has been clarified.

Ln 246. Looking at Fig 4 small phytoplankton appear to dominate in the North (> 52◦N). Although to a lesser extent than in the South. If correct, this sentence should be amended. Thank you, we have amended this sentence to read "On the mean, in the open waters of the North Atlantic, large phytoplankton have a greater contribution to the total biomass in the north and west (Fig 4a), but small phytoplankton are dominant to biomass throughout the basin and particularly in the south and east (Fig 4b)."

Ln 269. To say "only 40% of total biomass" seems strange. We have modified this to state "a smaller portion (40%)".

Ln 309. I would change the word "collaborative". Thank you. We feel the term is useful and elect to keep it.

Ln 361-364. This seems more suitable to the discussion than the results. Thank you. We find this a good lead in to the next line where the Discussion begins.

Ln 367. Type error. Remove "in" or "since". Thank you, fixed.

Ln 395. Type error. The use of "a" smooth climatological nutrient. . . Thank you, fixed.

Ln 409. Type error. . . ."on" the edges of. . . Thank you, fixed.

Ln 414-415. Suggest improving sentence readability. Perhaps: . . ."with the dominant mechanisms shifting across timescales" Thank you, fixed as suggested.

Ln 418. It is not clear to me what is meant by a "granular approach". Thank you, we have clarified to state "smaller subregions".

Ln 419. Type error. . . ."in" this region.  Thank you, fixed.

Ln 425. Type error. . . ."in the" northwest region.  Thank you, fixed.

Ln 434. Type error. "of" value or "valuable" Thank you, fixed.

Ln 443. Type error. Remove "in". Thank you, fixed.

References Laufkötter, C., Vogt, M., Gruber, N., Aita-Noguchi, M., Aumont, O., Bopp, L., et al. (2015). Drivers and uncertainties of future global marine primary production in marine ecosystem models. Biogeosciences, 12(23), 6955–6984. https://doi.org/10.5194/bg- 12-6955-2015 Kwiatkowski, L., Bopp, L., Aumont, O., Ciais, P., Cox, P. M., Laufkötter, C., et al. (2017). Emergent constraints on projections of declining primary production in the tropical oceans. Nature Climate Change. https://doi.org/10.1038/nclimate3265

Figure S2

[Figure]

**(a) Zooplankton Biomass (mgC m⁻³)**
**1998-2007, 0-100m Mean**

**(b) Zooplankton Biomass (mgC m⁻³ y⁻¹)**
**1998-2007 Trend**

**Fig. 1.** new figure S2

**Figure S4**

**Barotropic Streamfunction (Sv yr⁻¹)**
**1998-2007 Trend**

**Fig. 2.** new figure S4

---

## Author Comment (AC2) · 9 May 2018

RESPONSE to Anonymous Referee #2

Summary The manuscript integrates satellite ocean color observations and a coupled ocean physical-biogeochemical model to quantify plankton trends in the temperate and subpolar North Atlantic and evaluate potential underlying mechanisms. The model is an essential component for identifying physical transport effects on plankton dynamics. The study builds on a substantial literature on this important scientific question.

Thank you.

Overall, I found the manuscript to be relatively weak. The modeling study focuses on linear trends that are likely not robust over such a small time window (1998-2007) for

[Figure]

SeaWiFS data. The trend analysis leaves out the longer integrated SeaWiFS-MODIS data set, which exhibits substantial interannual to decadal variability, often with different temporal patterns than inferred from the shorter linear trend analysis. The numerical modelling is also limited to 2009 and thus is not compared with later MODIS data.

Thank you for your comments that have helped us to strengthen the manuscript. We describe below in more detail why we focus on linear trends. We do compare to MODIS data in Figure 2.

ocean and is a significant contribution that fits well within the scope of Biogeosciences journal. The manuscript is generally well written.

Thank you for this positive assessment.

Methodology The manuscript utilizes well-documented satellite ocean color data from SeaWiFS and MODIS-Aqua and an established ocean physical-biogeochemical model and hindcasting techniques. The study utilizes a set of monthly diagnostics for the physical and biological terms affecting the phosphate budget. The biological diagnostics are available for the SeaWiFS period (1998-onward) but not prior to 1998 (model output lost; Line 135). This subtracts some from the trend analysis over longer time periods 1949-2009 where only model biomass is available. Also, it is not clear why the model hindcast stops in 2009, now more than 8 years ago.

The reasons for the hindcast ending are found in the original manuscript on Lines 123-125. While, this is unfortunate, the model does cover the prime viewing period of SeaWiFS whose trends we aim to explain and thus it is a useful tool for the desired purpose.

Results One limitation with the analysis is the focus on linear trends over a relatively short analysis window (1998-2007) (Figure 1). As is clearly shown by Figure 2, the regional temporal patterns primarily exhibit inter-annual to decadal trends and any linear trend is relatively small and sensitive to the choice of time window (an issue that

the lead author is well familiar with and has published on previously). For example, the manuscript identifies declining biomass east of 30-35 deg. W (e.g. Line 11-12; Line 201-202), however, this is not consistent with the observations. The SeaWiFS-era trends in (Figure 1c) show only a small region of declining trends in the northeast (north of 55 deg.) with a substantial region of positive trends (though admittedly not statistically significant). The regional trends in Figure 2 that include MODIS data do not show such clear trends, and in fact from the merged SeaWiFS-MODIS data the trend in the southeast actually change signs.

We sense that the reviewer is concerned that our focus on trends implies a focus on long-term trend, perhaps even the suggestion of climate-change driven trends. This is most definitely not what we imply. These are trends over a specific period, 1998-2007, as observed by SeaWiFS – but they are in the context of interannual variability, as highlighted explicitly in the manuscript by the timeseries correlation analysis for the full model experiment (1949-2009). In the southeast (Figure 2a), the MODIS does indeed change sign, but this occurs after 2010 which is beyond our prime analysis period, and thus there is no inconsistency between the model, SeaWiFS and MODIS for the 1998-2007 period as suggested by this reviewer comment.

The manuscript would be much stronger if the focus was expanded to include the agreement (or disagreement) of model and observed interannual variability and underlying mechanisms. At a minimum, there needs to be more up front discussion of the rationale for and limitations of focusing on linear trends.

We show clearly with the figure several figures that the model does agree well with the satellite-observed biomass trends, and also reference previous manuscripts that have shown model fidelity against other datasets. Further, the analysis presented in the manuscript is precisely of the mechanisms driving these interannual changes, as asked for by the reviewer with this comment. Clearly, there is a need to clarify for the reader. Thus, we add text to the end of introduction that clarifies that this is a mechanistic analysis of the drivers of a particular set of SeaWiFS-observed changes in

biomass – which are best quantified as linear trends given the 10 year period available to us. This mechanistic analysis is combined with an effort to understand the degree to which these drivers are responsible for variability across the full model experiment. We also contrast this analysis to others that could be done based on the primary modes of variability across many decades, such as using an Empirical Orthogonal Function (EOFs) – work of the type that this author team has published extensively. The negative of EOF-type analysis is that at best, it tends to explain at most 30% of the large-scale variance over timeframes longer than most datasets – thus EOFs do not fully explain the observations. This paper is a case study of a particular period in which we are able to fully explain the drivers of the observed changes as estimated using a reasonable modeling tool (Figure 9,10).

The singular focus on annual means in the model and data analysis neglects the substantial seasonality in bloom dynamics in the subpolar North Atlantic. This raises two issues. First, there is no discussion of the robustness of annual mean satellite observations because of sampling biases, particularly during winter. Second, it is not clear if annual mean biomass is the biologically most important indicator; would more relevant indicators be peak surface biomass concentration or peak integrated biomass (ala arguments of Berhenfeld and Boss). Implicitly the analysis also assumes that only bottom-up factors (light and nutrient limitation) influence trends in phytoplankton biomass, neglecting possible top-down factors. This may be true for the model, but perhaps is an incomplete picture of the actual ocean.

We agree that alternative choices could have been made in the presentation of the biomass – peak vs. annual mean, for example. What is actually critical is that we are consistent between treatment of the observations and of the model, as we are. In response also to Reviewer 1, we add a figure of zooplankton biomass trends to the supplementary. This figure shows that top-down drivers are not driving the changes in this model. As discussed, this certainly does not rule out top-down drivers being important in the real ocean, but they are not required to capture the observed changes

in phytoplankton biomass.

Specific comments Line 180: I have some concerns regarding the following paragraph: "This analysis is based on annual mean fields. A 3-month lag of the biology diagnostics and biomass fields after physical diagnostics and other physical fields is employed to account for the maximum physical forcing occurring in the winter prior to the spring bloom. Thus, annual mean physical fields are averaged from October of the prior year to September of the year in question. Biological fields are January to December averages." I understand the need perhaps to adjust the year window to capture the relevant Fall and early winter pre-conditioning of subsequent spring bloom, but the text is not framed in terms of pre-conditioning. Rather a somewhat arbitrary 3-month lag is argued, inconsistent with the well-observed latitudinal seasonal patterns in the timing or phenology of bloom dynamics for the North Atlantic. Further, it is not clear that the relevant physical quantities are annual means for variable such as mixed layer depth with large seasonal variation and where it is more likely that the maximum winter mixed layer depth is more biologically relevant. Given the richness of monthly model output, a more nuanced data analysis would be warranted.

We add mention that a lag of 2 or 4 months does not substantially change results. Results are also similar with 0 lag or 1 month lag, but correlations are weaker. Given that our focus is on the northern North Atlantic, north of 40N, where deep mixing precedes the bloom by several months, some temporal lag is reasonable when annual means are being considered. Again, the use of annual means is a choice that must be made early in the analysis. As stated above, what is critical is that we aim to explain annual mean changes in SeaWiFS observed biomass, and that we do so with annual mean changes in the model. We agree that an analysis of also of monthly fields could be interesting, it is beyond the scope of the work already presented here. We add this suggestion in the last paragraph of the Discussion.

Line 209-212 "In both observations and models, the magnitudes of these changes are large in comparison to the mean. In the declining regions, where mean biomass is 15-

25 mgC m-3 (Fig. 1, S1), trends -0.5 to -1.5 mgC m-3 yr-1 over 10 years imply biomass reductions of 30-50 %. To the west of 30-35 deg. W, increases of a comparable percentage are implied." The magnitude of these trends may be appropriate for some pixel level trends but the magnitude of the trends are roughly an order of magnitude smaller at the regional scale in Figure 2.

Thank you for noting the need for clarification. We now include region-mean percent changes in the text: for model (seawifs): -17% (-19%) in SE, -10% (-15%) in NE, +9% (6%) in NW. The changes are based on the same model output and data shown in figures 2 and S1.

Figures 1 and 2; Lines 213-219: Regional analysis boxes are identified for the model in Figure 1d and linked to the time-series in Figure 2. It appears from the text that the same regional boxes are used for model and satellite observations, but it is unclear if this is appropriate given the spatial mismatch in the model and observed mean and trend patterns. The text (Lines 226-229) argue that this has a minimal effect but this should probably be shown in some figures in the supplement.

Thank you for suggesting we take another look at this. The differences are very minor, and thus additional supplemental figures would only be confusing. For example, this comparison of the NE box with MODIS and SeaWiFS boxes shifted to the north and east by 5degrees is essentially indistinguishable from Figure 2b without the shift.

Line 237-239: There is considerable nutrient data (though still sparse) from the CLIVAR Repeat Hydrography for the sub-polar North Atlantic, particularly from the German and Canadian occupied lines; worth looking in to.

Thank you – we look forward to seeing what the experts in analysis of these data find with respect to temporal changes in large-scale nutrient fields between the WOCE and CLIVAR eras given the spatial and temporal heterogeneity of these data. To perform this data analysis ourselves is clearly beyond the scope of this work.

Line 250: Are the annual means for phosphate and light limitation terms simply the straight means? Was there any consideration of weighting the limitation with seasonal variations in biomass or NPP, which might enhance the biological relevance.

As stated, the analysis is based on annual averages throughout for consistency. These limitations terms are very relevant to the biology and are the primary mechanism explored, so there is no need to "enhance the biological relevance" as suggested. Any weighting undertaken would also require additional analysis choices and thus a simple annual mean for all fields is the most straightforward approach when our goal is to explain annual mean SeaWiFS biomass changes.

Line 285: Does the model mixed layer trend agree with observations?

Yes it does, we add reference to Vage et al. 2008. Thank you.

Line 296-299: The wording of the text here is awkward; would be phrased as, for example, "horizontal advective divergence" (or convergence), etc. As written, the text and figure labels (Figure 8 and 9) confuse "flux" with "flux divergence".

Thank you, we have clarified the text as "flux convergence" and "flux divergence" throughout.
* * *
[Figure]

**Fig. 1.** comparison figure for timeseries NE with shift

---

## Author Response (AR1)

**Columbia University**
IN THE CITY OF NEW YORK
LAMONT-DOHERTY EARTH OBSERVATORY

June 8, 2018

Dear Dr. Herndl –

My co-authors and I re-submit this manuscript, *Mechanisms of northern North Atlantic biomass variability*, manuscript number bg-2018-89. As requested, we have thoroughly revised the manuscript in accordance with the reviewer comments.

We have responded in full to the reviewer comments that we received on May 2, 2018. Our primary changes have been to (1) address the question of zooplankton impact on biomass trends, (2) consider the impact of alternative satellite algorithms on the observed trends, and (3) clarified in multiple points in the text that our use of trends is not intended to suggest long-term climate driven trends, but instead short-term changes consistent with internal variability. We take pains to clarify that this is a mechanistic study that identifies plausible mechanistic explanations for observed biomass changes over the prime viewing period of the SeaWiFS satellite. These changes, and other edits that were suggested, have substantially increased the manuscript's clarity.

Thank you in advance for your attention to this manuscript.

Sincerely,

Galen McKinley, Professor
845 365 8585 | mckinley@ldeo.columbia.edu

RESPONSE to **Anonymous Referee #1**

This manuscript explores some of the mechanisms controlling phytoplankton biomass variability in the North Atlantic over the later 20th century and early 2000s. The manuscript is interesting and well written, and to some extent appears to challenge the view that nutrient dependent biomass variability is controlled only by the vertical nutrient supply. However, I have some reservations with the method adopted and cannot therefore recommend immediate publication. My main issue with the manuscript is the approach of using correlation coefficients between biomass and light/phosphate limitation terms as a means of attributing causality. This seems to be something of a shortcut given that these factors are likely to be somewhat collinear with other potential drivers of phytoplankton biomass. If possible, I think a more complete approach would be to recompute the model phytoplankton biomass using the limitation terms and other drivers (similar to what is done in Laufkötter et al., 2015 for several biogeochemistry models). This would allow the authors to assess the separate impacts of bottom-up processes (the influence of limitation terms on growth rates) as well as top-down loss terms (mortality/grazing). If this approach is not possible due to a lack of model output then I think some of the paper's conclusions should be toned down especially when using these correlation coefficients to infer the processes driving SeaWiFS variability.

*We address this concern by plotting zooplankton trends in the model over the main analysis period (new Figure S2 and included here). This figure shows that zooplankton trends occur of the same sign as of biomass. Were top-down processes driving the declines (increases) in biomass, then one would expect to see increasing (decreasing) zooplankton trends in the southeast and northeast (northwest). This is not what is occurring in the model. Thus, it is clear that nutrient and light trends are the primary drivers for the modeled trends. We include mention of this analysis in the main text (section 3.2), with reference to this new Figure S2.*

Figure S2

[Figure]

My other issue is that a number of processes that could be responsible for some of the trends in biomass variability seem to be neglected. These are perhaps not included in the model but this should nonetheless be stated. What role does temperature play? Is umax independent of temperature? What about zooplankton grazing rates? If grazing is temperature dependent does this explain any of the biomass variability? These sorts of things may be important given that certain models seem to show phytoplankton biomass declines despite increases in phytoplankton growth rates, due to overwhelming increases in losses to zooplankton grazing (Laufkötter et al., 2015).

*Thank you. We add mention of the temperature dependence of growth in equation 1. Grazing is not temperature dependent and this is also now mentioned in section 4. In this model, bottom-up drivers from nutrient and light limitation are responsible for the trends. In the context of zooplankton grazing, to section 4 we add mention of Laufkötter et al., (2015), also noting that the difference of findings may be related also to the very different timeframes and levels of forcing for trends in that paper (~100 years), as opposed to this analysis (10 years). Details of these additions to the text are below.*

Specific comments

Ln 90. What was the decision behind the use of the CbPM algorithm? Given that alternative algorithms can substantially differ it would be good to know that the trends described are robust to this algorithm choice. Perhaps a supplementary figure could be produced comparing CbPM mean state and trends in this region with an alternative algorithm such as VGPM.

*Thank you. CbPM is the only algorithm of which we are aware the estimates biomass from satellite. Biomass is the best point of comparison to the model since it is directly carried in the model. Nonetheless, to reassure the reviewer, we add comparison to CbPM and VGPM net primary productivity with a new Figure S1. Though the VGPM does suggest much higher NPP, this algorithm is considered less reliable than CbPM based on the publications of Behrenfeld who was a key developer for both VGPM and CbPM (Behrenfeld et al. 2005). Though magnitudes differ, the two algorithms indicate similar patterns of NPP trends.*

Ln 107. I think more model details are needed here even if they are published else- where. Specifically, what is meant by a "phosphorus-based ecosystem"? It would be good to have some mention of N. Is everything assumed to be Redfield? If so, is this a potential limitation of using this sort of model in this context? Is there any N fixation in the model?

*Thank you. We clarify that the primary macronutrient in the model is phosphorus, and that silicate is also limiting to the large phytoplankton class. Iron is a micronutrient. Consistent with other lower-complexity ecosystem models, such as BLING (Gailbraith et al. 2010, Biogeoscience), there is no nitrogen in this*

*model. We add note of this in section 2.4.*

Ln 140-160. See general comments above. Where does temperature limitation fit in? If not at all then I think this should be mentioned. Also, this section focuses on the effects of limitation terms on growth rates yet the analysis focuses on biomass not growth rates. I think the authors could better describe how growth rates and biomass are related, mentioning the additional processes that affect biomass in their model (e.g. zooplankton grazing?).

*Thank you. As shown above, zooplankton grazing follows biomass changes but does not drive them. The fact that this model, though it does not have all the potential complexity it might, does capture quite well the observed changes in biomass over this period. The success of the model strongly suggests that additional complexity is not required to reasonably explain the observed changes. We note in section 3.2 that zooplankton are not the driver of biomass changes in this model, and discuss this finding in context of Laufkotter et al. 2015 in section 4.*

*To section 3.2, we add "Modeled anomalies are not due to zooplankton top-down pressure on biomass, as evidenced by zooplankton trends that are positively correlated with biomass trends (Fig. S2). Thus nutrient and light, the bottom-up drivers in this model that change in a manner that drives biomass changes consistent with observations (Fig. 1), are the focus of this analysis."*

*To section 4, we add "In the context of 21$^{st}$ century climate-driven changes in biomass, Laufkötter et al., (2015) find zooplankton grazing to be important to biomass in some models under a strong climate change forcing scenario (RCP8.5). Zooplankton is not the driver of biomass changes in this model (Figure S3), with the very different timescales and levels of forcing for change - 10 years of interannual variability in this study, ~100 years with strong forcing in Laufkötter et al. (2015) - likely being a factor in this difference. That zooplankton grazing is not temperature dependent in this model may also contribute, but any potential effects would be limited by the annual mean temperature change from 1998-2000 to 2005-2007 being substantially smaller (+0.02 $^{o}$C, +0.28 $^{o}$C, +0.13 $^{o}$C, in SE, NE and NW regions, respectively) than over the 21$^{st}$ century in the RCP8.5 scenario (1-4 $^{o}$C at 40-60$^{o}$N, Laufkötter et al. 2015).*

Ln 401-404. Although declines in the horizontal nutrient supply may be the proximate driver is the ultimate driver not declines in the vertical supply to the west of the SE box? If so, perhaps this statement should be more nuanced.

*Thank you. We have modified the text to read "to LOCALLY increased stratification" so as to clarify this point.*

Figs 6 and 7. The differences between panels a and b are difficult to see in these plots. It would be useful to add a panel to each of these figures that is the difference between these time slices.

*We agree that these are somewhat hard to see, but the difference plots are unfortunately not much easier to look at. We have included MLD changes in timeseries form already in Figure 7c-e, and have added the difference plot for barotropic streamfunction in Figure S4.*

Technical/minor corrections

Ln 25-28. I think some references are needed in this paragraph.

*Thank you, references to the text by Sarmiento and Gruber (2006) as well as Sverdrup 1953; Dutkiewicz et al. 2001; Follows and Dutkiewicz, 2002 have been added.*

Ln 38. Type error. ". . .do not fit their. . ."

*Thank you, this has been fixed.*

Ln 44-48. Within this context it might be useful to mention that Kwiatkowski et al., 2017 related interannual variability of productivity to long-term trends across an ESM ensemble. Capturing productivity variability may therefore help reduce long-term projection uncertainties.

*Thank you for the suggestion. While Kwiatkowski et al. 2017 is a very nice paper, it is focused on the tropical oceans. The discussion here is about the Sverdrup hypothesis, which is relevant to the subpolar oceans. We do not feel that adding this reference is appropriate.*

Ln 74-75. "substantial change" reads as if there has been a climatic shift in the North Atlantic subpolar gyre. I think the authors are only referring to variability here and should clarify this.

*Thank you. It is stated in the following sentence "There is evidence these changes occur in response to changing buoyancy forcing and wind stress, in turn associated with modes of climate variability,.." which clarifies the relationship to variability.*

Ln 101. Space before units for consistency. "2200 m"

*Thank you, this has been fixed.*

Ln 109. I think something should come after "small". Small phytoplankton or nanophy- toplankton?

*Thank you, this has been fixed.*

Ln 172. Space before units for consistency. "100 m"

*Thank you, this has been fixed.*

Ln 224. I don't think "all three timeseries" is correct looking at Fig 2a. MODIS does not appear to have any positive anomalies prior to 2004.

*Thank you, this has been clarified.*

Ln 246. Looking at Fig 4 small phytoplankton appear to dominate in the North (> 52°N). Although to a lesser extent than in the South. If correct, this sentence should be amended.

*Thank you, we have amended this sentence to read "On the mean, in the open waters of the North Atlantic, large phytoplankton have a greater contribution to the total biomass in the north and west (Fig 4a), but small phytoplankton are dominant to biomass throughout the basin and particularly in the south and east (Fig 4b)."*

Ln 269. To say "only 40% of total biomass" seems strange.

*We have modified this to state "a smaller portion (40%)". We also have updated the text to refer to Figure 4 where this smaller portion is illustrated.*

Ln 309. I would change the word "collaborative".

*Thank you. We have removed this term.*

Ln 361-364. This seems more suitable to the discussion than the results.

*Thank you. We find this a good lead in to the next line where the Discussion begins.*

Ln 367. Type error. Remove "in" or "since".

*Thank you, fixed by removing "in".*

Ln 395. Type error. The use of "a" smooth climatological nutrient. . .

*Thank you, fixed.*

Ln 409. Type error. . . ."on" the edges of. . .

*Thank you, fixed.*

Ln 414-415. Suggest improving sentence readability. Perhaps: . . ."with the dominant mechanisms shifting across timescales"

*Thank you, fixed as suggested.*

Ln 418. It is not clear to me what is meant by a "granular approach".

*Thank you, we have clarified to state "smaller subregions".*

Ln 419. Type error. . . ."in" this region.

*Thank you, fixed.*

Ln 425. Type error. . . ."in the" northwest region.

*Thank you, fixed.*

Ln 434. Type error. "of" value or "valuable"

*Thank you, fixed.*

Ln 443. Type error. Remove "in".

*Thank you, fixed.*

time window (an issue that the lead author is well familiar with and has published on previously). For example, the manuscript identifies declining biomass east of 30-35 deg. W (e.g. Line 11-12; Line 201-202), however, this is not consistent with the observations. The SeaWiFS- era trends in (Figure 1c) show only a small region of declining trends in the northeast (north of 55 deg.) with a substantial region of positive trends (though admittedly not statistically significant). The regional trends in Figure 2 that include MODIS data do not show such clear trends, and in fact from the merged SeaWiFS-MODIS data the trend in the southeast actually change signs.

*We sense that the reviewer is concerned that our focus on trends implies a focus on long-term trend, perhaps even the suggestion of climate-change driven trends. This is most definitely not what we imply. These are trends over a specific period, 1998-2007, as observed by SeaWiFS – but they are in the context of interannual variability, as highlighted explicitly in the manuscript by the timeseries correlation analysis for the full model experiment (1949-2009). In the southeast (Figure 2a), the MODIS does indeed change sign, but this occurs after 2010 which is beyond our prime analysis period, and thus there is no inconsistency between the model, SeaWiFS and MODIS for the 1998-2007 period as suggested by this reviewer comment.*

*To clarify to the reader that we make no implication about long-term trends, we add to the abstract the following: "These short-term changes, attributable to internal variability, offer an opportunity to explore the mechanisms of the coupled physical-biogeochemical system. We use a regional biogeochemical model that captures the observed changes for this exploration."*

The manuscript would be much stronger if the focus was expanded to include the agreement (or disagreement) of model and observed interannual variability and underlying mechanisms. At a minimum, there needs to be more up front discussion of the rationale for and limitations of focusing on linear trends.

*We show clearly with the figure several figures that the model does agree well with the satellite-observed biomass trends, and also reference previous manuscripts that have shown model fidelity against other datasets. Further, the analysis presented in the manuscript is precisely of the mechanisms driving these interannual changes, as asked for by the reviewer with this comment. Clearly, there is a need to clarify for the reader. Thus, we add the above text to the abstract to clarify. At the end of section 1 we also add, "This is a mechanistic analysis of the drivers of SeaWiFS-observed changes in biomass that are best quantified as linear trends given the 10 year prime observational period. The degree to which these drivers are responsible for internal variability across the full model experiment (1948-2009) is also explored. Our approach can be contrasted to other possible approaches such as the use of Empirical Orthogonal Function (EOFs) to consider dominant modes of variability [Ullman et al. 2009; Breeden and McKinley 2016]. The negative of EOF-type analysis is that it tends to explain at most 30% of the large-scale variance, and thus does not fully explain observations. This paper is a case study for a particular period in which we aim to explain the drivers of the observed changes as fully as possible*

*using a model that represents well the observed changes. .”*

The singular focus on annual means in the model and data analysis neglects the substantial seasonality in bloom dynamics in the subpolar North Atlantic. This raises two issues. First, there is no discussion of the robustness of annual mean satellite observations because of sampling biases, particularly during winter. Second, it is not clear if annual mean biomass is the biologically most important indicator; would more relevant indicators be peak surface biomass concentration or peak integrated biomass (ala arguments of Berhenfeld and Boss). Implicitly the analysis also assumes that only bottom-up factors (light and nutrient limitation) influence trends in phytoplankton biomass, neglecting possible top-down factors. This may be true for the model, but perhaps is an incomplete picture of the actual ocean.

*We agree that alternative choices could have been made in the presentation of the biomass – peak vs. annual mean, for example. What is actually critical is that we are consistent between treatment of the observations and of the model, as we are. In response also to Reviewer 1, we add a figure of zooplankton biomass trends to the supplementary. This figure shows that top-down drivers are not driving the changes in this model. As discussed, this certainly does not rule out top-down drivers being important in the real ocean, but they are not required to capture the observed changes in phytoplankton biomass. To section 3.2, we add “Modeled anomalies are not due to zooplankton top-down pressure on biomass, as evidenced by zooplankton trends that are positively correlated with biomass trends (Fig. S2). Thus nutrient and light, the bottom-up drivers in this model that change in a manner that drives biomass changes consistent with observations (Fig. 1), are the focus of this analysis.”*

Specific comments Line 180: I have some concerns regarding the following paragraph: “This analysis is based on annual mean fields. A 3-month lag of the biology diagnostics and biomass fields after physical diagnostics and other physical fields is employed to account for the maximum physical forcing occurring in the winter prior to the spring bloom. Thus, annual mean physical fields are averaged from October of the prior year to September of the year in question. Biological fields are January to December averages.” I understand the need perhaps to adjust the year window to capture the relevant Fall and early winter pre-conditioning of subsequent spring bloom, but the text is not framed in terms of pre-conditioning. Rather a somewhat arbitrary 3-month lag is argued, inconsistent with the well-observed latitudinal seasonal patterns in the timing or phenology of bloom dynamics for the North Atlantic. Further, it is not clear that the relevant physical quantities are annual means for variable such as mixed layer depth with large seasonal variation and where it is more likely that the maximum winter mixed layer depth is more biologically relevant. Given the richness of monthly model output, a more nuanced data analysis would be warranted.

*We add mention to section 2.5 that a lag of 0 to 4 months does not substantially change results though correlations are weaker. Given that our focus is on the northern North Atlantic, north of 40N, where deep mixing precedes the bloom by several months, some temporal lag is reasonable when annual means are*

*being considered. Again, the use of annual means is a choice that must be made early in the analysis. As stated above, what is critical is that we aim to explain annual mean changes in SeaWiFS observed biomass, and that we do so with annual mean changes in the model. We agree that an analysis of also of monthly fields could be interesting, it is beyond the scope of the work already presented here. We add this suggestion in the last paragraph of the Discussion: "In order to address the simplest measure of change, we use annual mean fields for both the observations and the model. A deeper consideration of how these changes operate in the context of the significant seasonality of the region would be very interesting."*

Line 209-212 "In both observations and models, the magnitudes of these changes are large in comparison to the mean. In the declining regions, where mean biomass is 15- 25 mgC m-3 (Fig. 1, S1), trends -0.5 to -1.5 mgC m-3 yr-1 over 10 years imply biomass reductions of 30-50 %. To the west of 30-35 deg. W, increases of a comparable percentage are implied." The magnitude of these trends may be appropriate for some pixel level trends but the magnitude of the trends are roughly an order of magnitude smaller at the regional scale in Figure 2.

*Thank you for noting the need for clarification. We now include region-mean percent changes in the text. The changes are based on the same model output and data shown in figures 2 and S1. We now state "Regional mean changes in biomass from SeaWiFS (in the model) are -19% (-17%) in the SE region and -15% (-10%) in the NE. To the west of 30-35 °W in the NW region, regional mean changes are +6% (+9%)."*

Figures 1 and 2; Lines 213-219: Regional analysis boxes are identified for the model in Figure 1d and linked to the time-series in Figure 2. It appears from the text that the same regional boxes are used for model and satellite observations, but it is unclear if this is appropriate given the spatial mismatch in the model and observed mean and trend patterns. The text (Lines 226-229) argue that this has a minimal effect but this should probably be shown in some figures in the supplement.

*Thank you for suggesting we take another look at this. As we have stated in the text, the differences are very minor, and thus additional supplemental figures would only be confusing. For example, this comparison of the NE box with MODIS and SeaWiFS boxes shifted to the north and east by 5degrees is essentially indistinguishable from Figure 2b without the shift.*

[Figure]

Line 237-239: There is considerable nutrient data (though still sparse) from the CLIVAR Repeat Hydrography for the sub-polar North Atlantic, particularly from the German and Canadian occupied lines; worth looking in to.

*Thank you – we look forward to seeing what the experts in analysis of these data find with respect to temporal changes in large-scale nutrient fields between the WOCE and CLIVAR eras given the spatial and temporal heterogeneity of these data. To perform this data analysis ourselves is clearly beyond the scope of this work.*

Line 250: Are the annual means for phosphate and light limitation terms simply the straight means? Was there any consideration of weighting the limitation with seasonal variations in biomass or NPP, which might enhance the biological relevance.

*As stated, the analysis is based on annual averages throughout for consistency. These limitations terms are very relevant to the biology and are the primary mechanism explored, so there is no need to "enhance the biological relevance" as suggested. Any weighting undertaken would also require additional analysis choices and thus a simple annual mean for all fields is the most straightforward approach when our goal is to explain annual mean SeaWiFS biomass changes.*

Line 285: Does the model mixed layer trend agree with observations?

*Yes it does, we add reference to Vage et al. 2008. Thank you.*

Line 296-299: The wording of the text here is awkward; would be phrased as, for example, "horizontal advective divergence" (or convergence), etc. As written, the text and figure labels (Figure 8 and 9) confuse "flux" with "flux divergence".

*Thank you, we have clarified the text as "flux convergence" and "flux divergence" throughout section 3.5 and in the figures.*

[revised manuscript text omitted]

Figure 2

**Biomass Annual Anomalies (mgC m⁻³)**

[Figure]

(a) Southeast

(b) Northeast

[Figure]

(c) Northwest

Figure 3

[Figure]

Figure 4

[Figure]

Figure 5

[Figure]

Figure 6

[Figure]

(a) Barotropic Streamfunction (Sv)
1998-2000 Mean

(b) Barotropic Streamfunction (Sv)
2005-2007 Mean

Figure 7

[Figure]

Figure 8

[Figure]

(a) Vertical Phosphate Flux Convergence (mmol m$^{-3}$ yr$^{-1}$)
1998-2007, 0-100m Mean

(c) Total Physical Flux Convergence (mmol m$^{-3}$ yr$^{-1}$)
1998-2007, 0-100m Mean

(b) Horizontal Phosphate Flux Convergence (mmol m$^{-3}$ yr$^{-1}$)
1998-2007, 0-100m Mean

(d) Biological Phosphate Flux Convergence (mmol m$^{-3}$ yr$^{-1}$)
1998-2007, 0-100m Mean

[Figure]

Figure 9

[Figure]

Figure 10

[Figure]

[Figure]

---

## Author Response (AR2)

RESPONSE to **Referee #1 (Scott Doney)**

The study uses a coupled ocean biogeochemical-physical model to diagnose the patterns and inter-annual variability in upper-ocean (surface layer) phytoplankton biomass concentration in the north North Atlantic. After a brief model-data comparison against satellite ocean color data (SeaWiFS and MODIS), the study quantifies model dynamics in terms of small and large phytoplankton biomass, nutrient and light limitation terms, and the physical and biological divergence terms for surface phosphate. A central question in the study is the relative role of lateral versus vertical processes in governing regional biomass patterns in the western and eastern subpolar gyre.

Overall, the study is well constructed and presented. Disentangling causes of interannual variability is no simple task, even in a 3-D model, and the study provides a convincing case that lateral nutrient transport plays a role in regional biogeochemical variability.

After addressing relatively minor comments below, I feel the manuscript is suitable for publication.

*Thank you for the positive assessment. We have fully addressed these helpful comments, as detailed below.*

Specific Comments

As a general comment, it would be good to clarify in the analysis when a finding reflects model results versus field and satellite observations with caveats like "In the simulation ..." or "simulated large phytoplankton ..."

*We have added this text for clarity in multiple locations. Thank you.*

Line                                                                                                   25:
"Surface    ocean    phytoplankton    are    the    single    largest    biomass    pool    on    Earth"
The surface ocean phytoplankton biomass pool amounts to 1 to 2 Pg C. Terrestrial plant vegetation is several hundred Pg C. I do not think this statement is well supported and needs to be modified.

*Thank you for catching this error. We have corrected this, and the phrase now reads "Surface ocean phytoplankton contribute 50% of global net primary productivity [Field et al. 1998],…"*

Line 33: There has been considerable discussion in the literature over the past decade about Sverdrup's critical depth hypothesis and it's underlying assumptions (e.g., papers by Behrenfeld, Boss, Sosik, Mahadevan, etc.). Much of this discussion has been framed for the subpolar North Atlantic, and the Introduction would benefit from an expanded discussion on the interaction of light and mixing.

*Please see response to the next comment.*

Line 52: "that interannual variability in stratification is uncorrelated with that of productivity" On the other hand, Behrenfeld's work does show a relationship between winter mixing and biomass inventory. To some extent, this may reflect differences between surface layer concentrations and vertically-integrated inventories, a topic that could be addressed in the Discussion Section. The statement on line 57 is certainly true: "Thus, there is growing evidence that vertical processes alone are not sufficient to explain productivity variability." But the Introduction seems to underplay work by other research teams showing the importance of local vertical mixing processes. Even the results of this modeling study do show an influence for vertical mixing variability. The paper would benefit from a more balanced presentation.

*We have added substantially to the Introduction (lines 43-70 of the revised text) so that this body of work is appropriately represented. Thank you.*

Line 185: "Light limitation is calculated monthly in this way and then annually averaged." Was biomass-weighting considered when averaging monthly light and nutrient limitation terms to an annual value?

*Biomass weighting is not applied. We include note of this (line 201 of revision). Thank you.*

Line 239: "In these three regions, annual anomalies of biomass are compared to satellite observations from SeaWiFS            (1998-2007)            and            MODIS            (2003-2015)            (Fig.            2)." The paper would benefit from more quantitative analysis of observed-model comparisons. For example, what is the correlation and regression slope comparing annual anomalies between the model and the combined observational record?

*We have added comparison of slopes and correlations (lines 258-274 of revision). Thank you.*

Line 258: "The model captures the mean gradients of the phosphate field well, but mean values are 10-20 %            too            low            across            most            of            the            subpolar            gyre" Figure 3 shows surface concentrations, not horizontal gradients, so the reader must do their own gradient estimates in their head. A 10% difference in gradient is not easy to determine by eye. It would be good to match the analysis in the text to the figures; if lateral gradients are essential to the analysis, then gradients should be computed and shown explicitly. Also, this addresses the magnitude; presumably it is the vector interaction of the flow field with nutrient gradient vector that is key.

*We clarify here (line 278 of revised) that it is the large-scale pattern of phosphate that we intend to compare with Figure 3. Thank you.*

[revised manuscript text omitted]

Figure 2

[Figure]

920

Figure 3

[Figure]

Figure 4

[Figure]

**(a) Large Phytoplankton Biomass (mgC m⁻³)**
**1998-2007, 0-100m Mean**

**(c) Large Phytoplankton Biomass (mgC m⁻³ yr⁻¹)**
**1998-2007 Trend**

**(b) Small Phytoplankton Biomass (mgC m⁻³)**
**1998-2007, 0-100m Mean**

**(d) Small Phytoplankton Biomass (mgC m⁻³ yr⁻¹)**
**1998-2007 Trend**

Figure 5

[Figure]

Figure 6

[Figure]

Figure 7

[Figure]

Figure 8

[Figure]

Figure 9

**Phosphate Fluxes (mmol m$^{-3}$ yr$^{-1}$)**

[Figure]

[Figure]

[Figure]

Figure 10

[Figure]

(a) Vertical Phosphate Flux Convergence (mmol m$^{-3}$ yr$^{-2}$)
1998-2007 Trend

(c) Physical Phosphate Flux Convergence (mmol m$^{-3}$ yr$^{-2}$)
1998-2007 Trend

(b) Horizontal Phosphate Flux Convergence (mmol m$^{-3}$ yr$^{-2}$)
1998-2007 Trend

(d) Biological Phosphate Flux Convergence (mmol m$^{-3}$ yr$^{-2}$)
1998-2007 Trend